# How Much of a Pixel Needs to Burn to Be Detected by Satellites? A Spectral Modeling Experiment Based on Ecosystem Data from Yellowstone National Park, USA

**Mats Riet \* and Sander Veraverbeke** 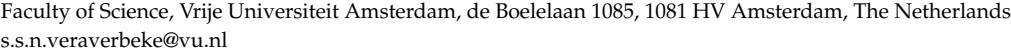

Faculty of Science, Vrije Universiteit Amsterdam, de Boelelaan 1085, 1081 HV Amsterdam, The Netherlands; s.s.n.veraverbeke@vu.nl
\* Correspondence: matsriet@xs4all.nl

**Abstract:** We present a simple modeling technique based on linear spectral mixture analysis to assess satellite detectability of sub-pixel burned area. Pixel observations are modeled using a linear combination of pure land covers, called endmembers. We executed an experiment using spectral data from Yellowstone National Park, USA. Using endmember samples from spectral libraries, pixel samples were assessed on burn detectability using the widely used differenced Normalized Burn Ratio (dNBR). While individual samples yielded differing results for Landsat 8, Sentinel-2, and the Moderate Resolution Imaging Spectroradiometer (MODIS), the average park-wide detectability of burned area was consistent across satellites. For the commonly used dNBR threshold of 0.15, the results indicated that detectability is reached when around a quarter of a pixel's area is burned. However, a significant percentage of the modeled burned pixels remained undetectable, especially those with low pre-fire vegetation cover. This has consequences for burned area estimates, as smaller fires in sparsely vegetated terrain may remain undetected in moderate resolution burned area products.

**Keywords:** burned area detection; differenced Normalized Burn Ratio; spectral mixture analysis; Yellowstone National Park

## 1. Introduction

Wildfires have severe impacts on the natural environment and human lives. They accelerate changes in ecosystems by removing vegetation [1,2], and, as such, can exacerbate soil erosion [3,4]. Furthermore, the release of large amounts of greenhouse gases from fires can contribute to climate warming when ecosystems do not fully recover [5]. In addition, air pollution from the fire's smoke can be deadly [6,7], and the economic costs of both prevention and damage repair are substantial [8]. For these reasons, monitoring fires is extremely important.

Satellite remote sensing is the only practical means of gathering information on fires over large areas. Two remote sensing techniques are often used to detect fires. Firstly, active fires can be detected by the thermal anomalies from their heat release [9]. Secondly, burned areas can be detected based on surface reflectance changes after the fire [10,11]. Fires remove vegetation, produce charcoal, and can expose soil. Active fires can only be detected during the fire, and detection accuracy thus depends on orbital characteristics of the satellites. In contrast, surface reflectance can be mapped after the fire from cloud-free images. This method is thus more suitable to map the extent of fires [12].

There are, however, also limits to the detection capabilities based on changes in surface reflectance. Image pixels, no matter how small, are always a composite of the constituents within them, and this sub-pixel heterogeneity can impact the analyses [13]. If only a small area within a pixel is burned, this may not be detectable by satellites, which can result in high omission errors [14].



An often-used metric for burned area detection (e.g., [15–19]), the Normalized Burn Ratio (NBR), may also suffer from this limitation. This metric takes advantage of spectral differences between reflectance of healthy vegetation and burned area in the near-infrared (NIR, ∼0.75–0.90 μm) and shortwave infrared (SWIR, ∼2.09–2.35 μm) domains of the electromagnetic spectrum [20,21]:

$$NBR = \frac{NIR - SWIR}{NIR + SWIR} \tag{1}$$

Healthy vegetation typically has high reflectance values in the NIR domain, but low values for SWIR. Reflectances for charcoal, soil, and bare rock are typically opposite of that, giving low NIR and high SWIR values in deforested or burned areas. As a result, a significant decrease in NBR is an indication of vegetation loss.

However, if only a small fraction of a pixel has burned, the NBR may not decrease much, leading to the burned area remaining undetectable. A large unknown here is how large the fraction of burned area needs to be before detection is possible. This results in some underestimation of fire impacts, as smaller contributions remain undetected. We aim to improve our understanding of this detection threshold by modeling the response of satellites to increasing burned fractions. To achieve this, the study area of Yellowstone National Park, USA, was selected to model spectral responses of burned area.

## 2. Materials and Methods

In this project, the linear version of spectral mixture analysis (SMA) was used to calculate satellite spectral response to burned areas. SMA is a widely used technique for modeling the composite nature of pixels [22]. For studies on wildfires, it has been used to estimate fire severity (e.g., [23,24]), intensity (e.g., [25]), and vegetation recovery (e.g., [26]). The technique is relatively simple, has been extensively tested, and has proven to be reliable [22]. The approach in SMA is to model the spectral signatures of a pixel as a mixture of pure land covers, called endmembers. A burned pixel may, for example, be a combination of the endmembers grass, bare soil, and charcoal. Typically, SMA is used to estimate abundances of such endmembers in pixels imaged by satellites. Here, we did not use satellite imagery; instead, we created artificial pixel samples from combining these endmembers and used these to determine satellite response. To our knowledge, no prior study has modeled satellite detectability of burned area in such a way.

Firstly, endmember samples were selected for the study area and obtained from spectral databases (see Section 2.2). Next, the samples were grouped and combined to model sub-pixel environments located within the park. A fire in these environments was subsequently modeled by introducing the spectral contribution of a charcoal sample. Finally, we assessed burned area detectability of our modeled spectral mixtures.

### 2.1. Study Area

The study area for which endmembers were selected is Yellowstone National Park (shortened here to Yellowstone), a nature preserve in Wyoming, USA [27]. The park covers 8903 km$^2$, and is a landscape characterized by high volcanic plateaus eroded by glaciers and rivers and flanked by mountains. Lakes cover around 5% of the park's area. Elevation ranges between 1637 m and 3512 m, averaging at 2479 m [28]. The area is volcanically active and contains three large calderas [29]. The park was selected as a study area because the available vegetation spectral data are complete and easily accessible. Additionally, wildfires occur often in the park during the summer season. The burned environments for this region were modeled using a linear combination of three endmembers: vegetation, substrate, and charcoal.

The dominant vegetation community in Yellowstone is coniferous forest, which represents around 85% of the surface area. The United States National Park Service classifies them under four communities. Of them, forests dominated by lodgepole pine (*Pinus contorta* subsp. *latifolia*) are by far the most common. In addition, there are forests dominated by spruce fir (*Picea engelmannii* and *Abies lasiocarpa*), whitebark pine (*Pinus albicaulis*), and douglas fir

(*Pseudotsuga menziesii*). This last vegetation community is interesting as it has the highest frequency of fire [27]. The vegetation patterns in the region vary greatly with elevation and topography, with douglas fir being located at lower elevations, lodgepole pine between 2000 and 2400 m, and the other communities reaching to the upper tree line [30]. Other vegetation communities in the park are grasslands, meadows, sagebrushes, and hydrothermal areas. These are classified under non-forested vegetation. Finally, scattered throughout the park are smaller communities of aspen forests, wetlands, and streamside vegetation [27].

*2.2. Endmember Selection*

The reflectance spectra of the environments in Yellowstone were modeled using three categories of endmembers: vegetation, substrate, and charcoal. Vegetation spectral data were obtained from the United States Geological Survey's (USGS) spectral library, which includes data collected from laboratory, field, and airborne spectrometers. Covering wavelengths between 0.2 and 200 μm, the library contains measurements of vegetation, minerals, chemical compounds, and manmade materials. Among this library, averaged spectral reflectance data of the dominant vegetation communities in Yellowstone National Park are available. These are averaged top-of-canopy measurements of vegetation communities, previously collected by the Airborne Visible/Infrared Imaging Spectrometer (AVIRIS) [31]. The used spectra span wavelengths between 370 and 2500 nm at an interval of 10 nm. The vegetation samples that were used in this study are summarized in Table 1. It is common in spectral mixture analysis to subdivide the vegetation endmember into two: green and non-photosynthetic parts of the vegetation cover. However, as the vegetation samples used are averages of plant communities, they represent spectra that contain both green and non-photosynthetic parts of the plants. Thus, the vegetation endmember samples from AVIRIS represent the vegetation spectra as they would be measured by a satellite instrument.

**Table 1.** Overview of vegetation samples used in the analysis. More detailed descriptions of the samples can be found in the USGS spectral library [31].

| Type | Vegetation Community | Dominant Species | No. of Samples |
|---|---|---|---|
| Forest | Douglas fir | *Pseudotsuga menziesii* | 3 |
| | Lodgepole pine | *Pinus contorta* subsp. *latifolia* | 11 |
| | Spruce fir | *Picea engelmannii, Abies lasiocarpa* | 5 |
| | Whitebark pine | *Pinus albicaulis* | 2 |
| Nonforest | Bacterial mat | *Chloroflexus aurantiacus, Synechococcus lividus* | 1 |
| | Conifer–meadow mix | | 1 |
| | Grass | *Festuca idahoensis* | 5 |
| | Sagebrush | *Artemisia tridentata* | 4 |
| | Sedge | | 1 |
| | Willow–sedge mix | | 1 |

The endmembers that were used to simulate the substrates in Yellowstone were obtained from the NASA Jet Propulsion Laboratory's ECOSTRESS spectral library. This library is a collection of laboratory measurements covering a wavelength range between 350 and 15,400 nm of (green and non-photosynthetic) vegetation, rocks, soils, minerals, and some manmade materials [32]. From this library, both rock and soil samples were obtained to simulate the substrate conditions of the park. Rock spectral samples were selected according to the geology of Yellowstone, as described by Keefer [33]. At the wavelengths used in this work (up to 2350 nm), the spectral samples from this library had a measurement interval of 1 nm.

The inclusion of soil samples is important, as some soil-forming processes influence the spectral signature. For example, increases in organic matter result in lower reflection in the NIR domain [34]. Furthermore, in the presence of charcoal, the Normalized Burn Ratio is highly sensitive to soil type [35]. However, selecting soils is less straightforward compared to rocks, as the characteristics of soils are highly variable. Apart from the lithology in which the soil is formed, it is highly dependent on the local climate characteristics [36]. Soils in

Yellowstone are typically in the frigid or cryic temperature regime, although warmer regimes form around hydrothermal areas. The climate of the park is characterized as moderately dry [28], and mostly falls within the udic moisture regime [36]. The most common soil types in the park are mollisols (thick, organic soils) and inceptisols (weakly developed mineral soils); however, entisols, andisols, and alfisols are present as well. For a full overview of the soils in Yellowstone, see Rodman et al. [28].

We only selected soils for the more common lithologies, as these have a higher impact on the park-wide results. Initially, the selected soils matched parent materials with the rocks in the park, could form in the temperature and moisture regimes of Yellowstone [36], and are similar to the soils described by Rodman et al. [28]. The rock and soil samples were grouped in accordance with the geological units, as shown in Table 2.

**Table 2.** Overview of substrate samples used in the analysis. More detailed descriptions of the samples can be found in the ECOSTRESS library [32].

| Geological Unit | Rock/Soil Name | No. of Samples |
|---|---|---|
| Precambrian Gneiss and Schist | Gneiss | 19 |
| | Schist | 26 |
| | Inceptisol dystrochrept | 1 |
| | Inceptisol haplumbrept | 1 |
| Paleozoic Formations | Dolomite | 3 |
| | Limestone | 33 |
| | Sandstone | 21 |
| | Shale | 25 |
| Mesozoic Formations | Limestone | 33 |
| | Sandstone | 21 |
| | Shale | 25 |
| Tertiary Formations | Conglomerate | 3 |
| | Sandstone | 21 |
| Diorite Intrusions | Diorite | 2 |
| | Granodiorite | 4 |
| Absaroka Volcanic Breccias | Andesite | 6 |
| | Basalt | 35 |
| | Mafic tuff | 1 |
| | Inceptisol haplumbrept | 1 |
| | Mollisol cryoboroll | 1 |
| Yellowstone Tuffs | Felsic tuff | 8 |
| | Alfisol fragiboralf | 1 |
| | Alfisol haplustalf | 2 |
| | Inceptisol xerumbrept | 1 |
| Plateau Rhyolite | Rhyolite | 24 |
| | Inceptisol cryumbrept | 1 |
| | Inceptisol plaggept | 1 |
| | Mollisol cryoboroll | 1 |
| Basalt Flows | Basalt | 35 |
| Quartenary Deposits | Travertine | 2 |
| | Alfisol fragiboralf | 1 |
| | Alfisol haplustalf | 2 |
| | Alfisol paleustalf | 1 |
| | Inceptisol cryumbrept | 1 |
| | Inceptisol haplumbrept | 1 |
| | Inceptisol plaggept | 1 |
| | Inceptisol xerumbrept | 1 |
| | Mollisol cryoboroll | 1 |
| | Mollisol haplustall | 1 |

However, some very common parent materials were greatly underrepresented in the soil samples due to a lack of soils that fit all the criteria. In these cases, some soils were chosen that are technically outside of this climatic regime, but fit well with the soil descriptions and parent material. Most notably, there was no soil sample found for volcanic ash that also fit the climatic regimes. Thus, the otherwise well-fitting "inceptisol xerumbrept" (which forms in deserts) was added.

The charcoal endmembers were collected by Veraverbeke et al. [37] from the 2011 Canyon fire scar in Kern County, California. Their spectra were measured at NASA's Jet Propulsion Laboratory, in the same fashion as the used rock and soil samples from their ECOSTRESS library. Three charcoal endmember samples were available and used.

### 2.3. Sample Preparation

The first step of the analysis was to calculate the near-infrared (NIR) and shortwave infrared (SWIR) response of the endmember samples (Figure 1). To perform this, we followed the procedure outlined by Barsi et al. [38]. For every wavelength of an endmember sample, the response of a satellite band was calculated:

$$\rho_{ems,b,\lambda} = \rho_{ems,\lambda} \cdot \beta_{b,\lambda} \tag{2}$$

where $\rho_{ems,b,\lambda}$ is the spectral reflectance of an endmember sample *ems* as observed by a satellite instrument band *b* at wavelength $\lambda$. $\rho_{ems,\lambda}$ is the spectral reflectance of the endmember sample from library (again at wavelength $\lambda$), and $\beta_{b,\lambda}$ is the spectral response of a satellite at that wavelength. Both vary between 0 and 1. The satellites for which detectability was assessed are Landsat 8, Sentinel-2, and the Moderate Resolution Imaging Spectroradiometer (MODIS) (Table 3). Their response functions are shown in Figure 2. The spectral response functions were linearly interpolated to match the spectral resolution of the endmember sample.

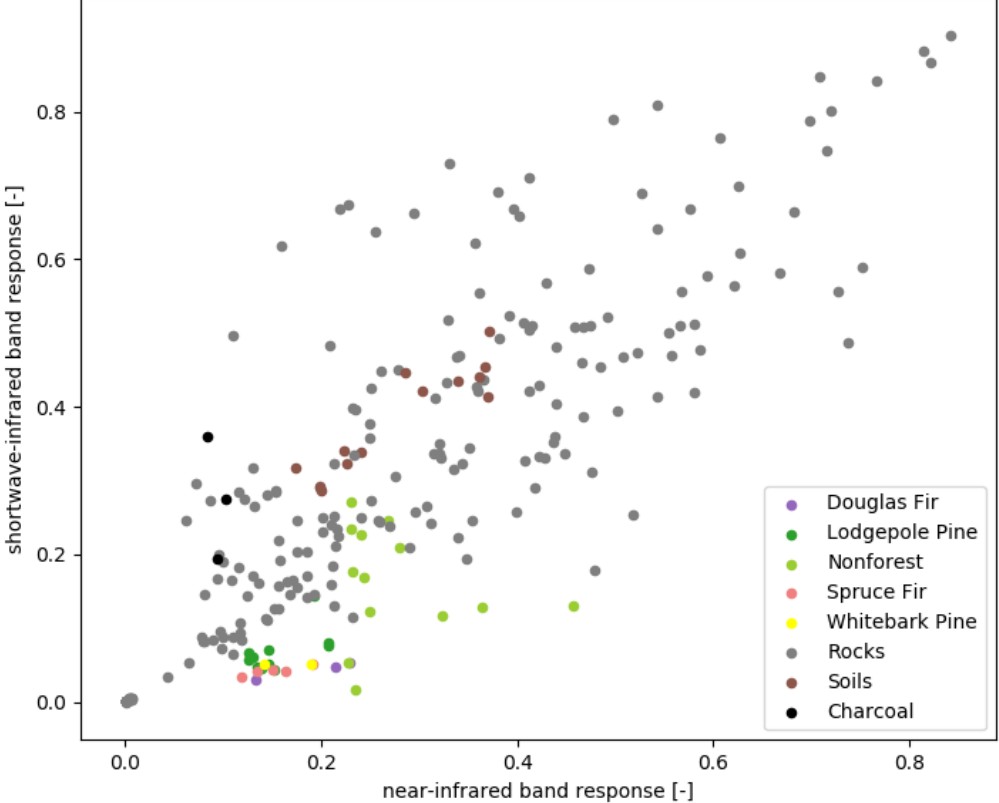

**Figure 1.** Near-infrared (NIR) and shortwave infrared (SWIR) band responses of Landsat 8 for the endmember samples.

**Table 3.** Details of the satellite instrument bands used in the analysis. Used are the near-infrared (NIR) and shortwave-infrared (SWIR) bands for Landsat 8, Sentinel-2 (two satellites; A and B), and the Moderate Resolution Imaging Spectroradiometer (MODIS, two satellites; Terra and Aqua).

| | NIR Band | | | SWIR Band | | | |
|---|---|---|---|---|---|---|---|
| Satellite Instrument | Band No. | λ Range (μm) | Resolution (m) | Band No. | λ Range (μm) | Resolution (m) | Data Source |
| Landsat 8 | 5 | 0.85–0.88 | 30 | 7 | 2.11–2.29 | 30 | [39] |
| Sentinel-2 A & B | 8 | 0.78–0.89 | 10 | 12 | 2.01–2.37 | 20 | [40] |
| MODIS Terra & Aqua | 2 | 0.84–0.89 | 250 | 7 | 2.11–2.16 | 500 | [41] |

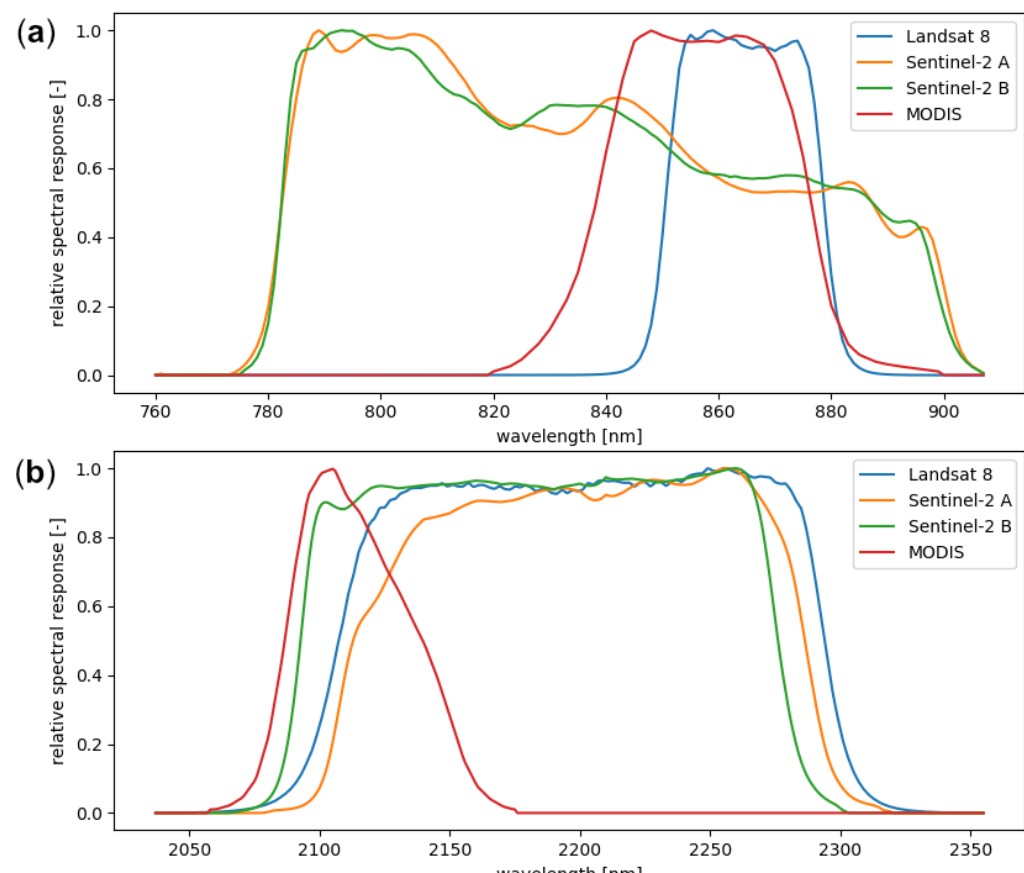

**Figure 2.** Instrument response functions of the instruments used: (**a**) Near-infrared (NIR) spectral bands. (**b**) Shortwave infrared (SWIR) spectral bands. The spectral response of the Sentinel-2 A and B instruments are slightly different, whereas the responses of the two Moderate Resolution Imaging Spectroradiometer instruments (MODIS Terra and Aqua) are identical.

The responses to these reflectances were then averaged over the band's wavelength range:

$$\rho_{ems,b} = \frac{\sum_{\lambda=\lambda_0}^{\lambda_1} \rho_{ems,b,\lambda} \cdot \Delta\lambda}{\sum_{\lambda=\lambda_0}^{\lambda_1} \beta_{b,\lambda} \cdot \Delta\lambda} \tag{3}$$

where $\rho_{ems,b}$ is the spectral response for the endmember sample *ems* of a given instrument band *b*, $\lambda_0$ and $\lambda_1$ are the wavelength boundaries of the instrument band, and $\Delta\lambda$ is the wavelength distance between the measurements of the endmember sample. In this case, multiplying the endmember reflectance with the satellite response functions automati-

cally sets the wavelength range (as the responses are set to 0 outside of the band of the instrument). If $\Delta\lambda$ is constant over the wavelength domain, Equation (3) can be simplified:

$$\rho_{ems,b} = \frac{\sum \rho_{ems,b,\lambda}}{\sum \beta_{b,\lambda}} = \frac{\sum \rho_{ems,\lambda} \cdot \beta_{b,\lambda}}{\sum \beta_{b,\lambda}} \tag{4}$$

For every endmember sample, Equation (4) was used to calculate the NIR and SWIR band values for the different satellite sensors.

### 2.4. Modeling Burned Area Response

In linear spectral mixture analysis, spectral endmembers are combined linearly to simulate the spectrum of a pixel [42]:

$$\rho_p = \sum_{ems=0}^{n} \rho_{ems} \cdot f_{ems} \tag{5}$$

Here, $\rho_p$ is the reflectance of a pixel, which is a weighted average of the reflectances of the endmembers used to simulate the pixel ($0 \leq \rho_{ems} \leq 1$), with weights given by their spectral contribution fractions ($0 \leq f_{ems} \leq 1$). In linear SMA, these fractions can also be interpreted to represent the surface area covers of the endmembers, and therefore are non-negative and have to add up to unity.

In this work, pixel spectra were modeled using three endmembers: vegetation, substrate, and charcoal. This yields the following calculation for the pixel reflectances:

$$\rho_p = \rho_v \cdot f_v + \rho_g \cdot f_g + \rho_c \cdot f_c \tag{6}$$

In this equation, $\rho_p$, $\rho_v$, $\rho_g$, and $\rho_c$ are the reflectance values of the pixel and its vegetation ($v$), substrate ($g$), and charcoal ($c$) endmembers, respectively. $f_v$, $f_g$, and $f_c$ are the respective cover contribution fractions. The letter $g$ (from ground) is used for substrate endmembers to disambiguate from the $s$ used for starting values in later equations.

The endmember reflectances in this equation (and further equations) are those obtained by Equation (4). Thus, the obtained pixel reflectance is for a given instrument band and is already scaled with the band's response function. This means that the NIR and SWIR values obtained from this equation are the reflectances of the pixel as would be obtained from the satellites.

Before a fire, a pixel only contains vegetation and substrate endmembers ($f_c = 0$); thus, Equation (6) yields

$$\rho_{p,s} = \rho_v \cdot f_{v,s} + \rho_g \cdot (1 - f_{v,s}) \tag{7}$$

$\rho_{p,s}$ is the pixel's ($p$) pre-fire (starting, $s$) reflectance. $f_{v,s}$ is the vegetation cover of the pixel sample before the fire. This value was varied between 0.05 and 1 in steps of 0.05, yielding 20 pixel samples per ground-vegetation endmember combination. The substrate endmember takes up the rest of the area of the pixel; therefore, its contribution fraction is equal to $1 - f_{v,s}$. From this pre-fire pixel reflectance, the Normalized Burn Ratio before fire ($NBR_{p,s}$) could be calculated:

$$NBR_{p,s} = \frac{\rho_{p,NIR,s} - \rho_{p,SWIR,s}}{\rho_{p,NIR,s} + \rho_{p,SWIR,s}} \tag{8}$$

The cover contributions for the pixels were then altered by replacing parts of the ground and vegetation with charcoal, simulating a burn in the pixel. The model calculates a decrease in the contribution of vegetation ($f_v$) using a variable fraction of burned vegetation ($f_b$). The fraction of burned vegetation ($0 \leq f_b \leq 1$) gives the fraction of the original vegetation ($f_{v,s}$) that is currently burned. The fraction of vegetation ($f_v$) could then be calculated using

$$f_v = f_{v,s} \cdot (1 - f_b) \tag{9}$$

Along with changes in $f_v$, the contribution of ground ($f_g$) and charcoal ($f_c$) also need to change, as the total fraction of the pixel needs to be 1. The parameter $\Delta c$ (or $\Delta$char) gives the change in charcoal cover fraction per unit of vegetation lost (burned). The remaining contributions could thus be calculated using

$$f_c = f_b \cdot f_{v,s} \cdot \Delta c \tag{10}$$

$$f_g = 1 - (f_v + f_c) \tag{11}$$

$\Delta c = 0$ represents vegetation loss without charcoal input. $\Delta c = 1$ represents a fire in which all vegetation is replaced by charcoal; the contribution of ground stays constant. $0 < \Delta c < 1$ represents an increase of both charcoal and bare ground. $\Delta c > 1$ indicates that part of the ground will be covered by charcoal in addition to vegetation loss. The parameter can not be negative, as this would result in an illogical mechanism where forest cover increases as it is burned. In the model, results were calculated for $\Delta c$'s between 0 and 1 in steps of 0.25.

Vegetation was removed until the vegetation contribution became 0, or the burned area became detectable. The detectability was assessed using the differenced Normalized Burn Ratio (dNBR), which is calculated by taking the difference between the pre- and post-fire NBR. After an increase in the burned vegetation fraction, the pixel's NBR ($NBR_p$) and, subsequently, dNBR were calculated:

$$NBR_p = \frac{\rho_{p,NIR} - \rho_{p,SWIR}}{\rho_{p,NIR} + \rho_{p,SWIR}} \tag{12}$$

$$dNBR = NBR_{p,s} - NBR_p \tag{13}$$

If the dNBR was greater than a certain threshold, the fire was set to be detectable and the cover fractions of the different endmembers $f_v$, $f_g$, and $f_c$ were saved. A burned pixel was set to be undetectable if the burned fraction reached 1 before the dNBR threshold was exceeded. In the model code, multiple threshold values were used to be able to assess the influence of this parameter. The dNBR thresholds varied between 0.05 and 0.25 in steps of 0.05.

The cover fractions were calculated for every substrate sample combined with every vegetation and charcoal sample. However, we were only interested in the range of these values for a certain combination, not necessarily the outputs for the individual library samples. For example, the combination of gneiss (19 samples) with lodgepole pine forest (11 samples) and charcoal (3 samples) has a total of $19 \times 11 \times 3 = 627$ model output values. However, the interest is only in the detectability of the gneiss–lodgepole pine combination. Therefore, of these 627 results, only the minimal, mean, and maximal values were saved. This reduced the number of results for this combination to three per variable.

From the burned fraction $f_b$, the endmember contributions can be calculated. The goal of the model was then to find the value of $f_b$ for which the burn could be detected. This can be achieved using a root-finding algorithm; for example, iteratively, by increasing stepwise $f_b$, testing the resulting dNBR and checking if it is above the threshold. Then $f_b$ could be tweaked to attain closer approximations of the threshold. However, it is possible to directly calculate the burned fraction needed for detection. The methodology is explained in Appendix A. The direct calculation is faster and more precise, allowing more samples to be processed within a reasonable amount of time.

### 2.5. Data Aggregation

With a high number of endmember sample combinations, aggregation of the results was required to draw conclusions. The results were aggregated at different levels (Table 4), allowing conclusions to be drawn at different scales.

**Table 4.** Result aggregation level details.

| Level | Name | Results Count | Notes |
|---|---|---|---|
| 0 | Endmembers | 49,572,000 | Results are not saved at this level. |
| 1 | Groupings | 1,170,000 | Contains min, mean, and max values of sample combination groupings. |
| 2 | Geologies | 300,000 | Substrate groupings aggregated to their geological units. |
| 3 | Park-wide | 6000 | Weighted average using abundances of geology–vegetation combinations. |

The direct result outputs are defined at level 0. At this level, the results are given per endmember sample combination; each one is calculated using a combination of one substrate, vegetation, and charcoal spectral sample.

As we are not interested in the variation between the samples of a particular endmember (e.g., between basalt samples), we can reduce the amount of data. We perform this by only saving some statistical properties of a collection of these sample combinations. Here, we save only the mean, minimal, and maximal values of the model results in a collection or grouping, essentially giving one value with uncertainty. This yields results at aggregation level 1. In this case, spectral samples are grouped according to their soil name, rock name (see Table 2), or vegetation community (see Table 1). Every grouping is then a combination of a named substrate with a vegetation community. For example, one of the grouping combinations is basalt with lodgepole pine, containing all model results that are a combination of basalt and lodgepole pine.

Higher aggregation levels than the groupings level require some information on the area covers of the substrate and vegetation combinations. However, the surface area covers of rocks or soils within a geological unit are largely not documented. For aggregation level 2, this is accounted for as follows. We assign every rock and soil to a geological unit (per Table 2). For the mean value of a unit, we assume the rock and soil types to be equally abundant on the surface. The high uncertainty in this assumption is taken into account by obtaining the minimal and maximal value of combinations assigned to a geological unit. These extreme values thus assume that the entire unit is dominated by the most extreme spectral samples within them, providing a constraint on the variability.

The final aggregation level, aggregation level 3, assesses burned area detectability at a park-wide scale. To achieve this, the results for the geological units were weighed with their surface area. To find these weights, a geological map [43] and vegetation habitat map [44] of Yellowstone National Park were reclassified and combined to show the geological unit–vegetation community combinations. Area sizes of these combinations were subsequently set as the weights in the result analysis. The classes and corresponding weights are shown in Table 5.

**Table 5.** Estimated area cover % of the lithology–vegetation combinations.

| | Douglas Fir | Lodgepole Pine | Spruce Fir | Whitebark Pine | Nonforest | Total |
|---|---|---|---|---|---|---|
| Precambrian Formations | 0.675 | 0.607 | 0.007 | 0.130 | 0.517 | 1.94 |
| Paleozoic Formations | 0.141 | 0.653 | 0.038 | 0.344 | 0.509 | 1.69 |
| Mesozoic Formations | 0.106 | 0.919 | 0.234 | 0.749 | 0.694 | 2.70 |
| Tertiary Formations | 0 | 0.002 | 0 | 0.001 | 0.001 | 0.003 |
| Diorite Intrusions | 0.081 | 0.079 | 0.005 | 0.148 | 0.236 | 0.549 |
| Absaroka Volcanics | 1.18 | 5.74 | 0.842 | 7.63 | 5.78 | 21.2 |
| Yellowstone Tuffs | 0.826 | 10.8 | 1.26 | 1.09 | 0.519 | 14.5 |
| Plateau Rhyolite | 0.073 | 18.8 | 1.74 | 2.14 | 1.75 | 24.5 |
| Basalt Flows | 0.230 | 1.39 | 0.045 | 0.039 | 0.145 | 1.86 |
| Quartenary Deposits | 1.11 | 13.4 | 6.30 | 1.97 | 8.25 | 31.0 |
| **total** | 4.42 | 52.5 | 10.5 | 14.2 | 18.4 | 100 |

At aggregation level 3, the model contains three results (min, mean, max) for the entire park. However, parameterization adds additional dimensions; thus, the park-wide results are dependent on a given dNBR threshold (5 options), $\Delta$char (5), instrument (4), and starting vegetation fractions $f_{v,s}$ (10), totaling 6000 result values.

## 3. Results

The outputs of the model vary depending on the parameter settings used. Thus, we analyze the detectability by assessing the influences of the parameters. For most of the results, the starting fraction of vegetation ($f_{v,s}$) is set as the independent variable, as detection characteristics are highly dependent on it.

### 3.1. Differences between Satellite Sensors

First, we identified the differences introduced by the various satellite instruments. We use the aggregation level 2 for this purpose, as it still contains all the variability of the lower aggregation levels while being attributable to certain areas of the park. The deviation in burned fraction ($f_b$) on detection is shown in Figure 3, illustrating how much vegetation loss is required before detection.

A few observations are of note here. Firstly, deviations are larger with increases in starting vegetation fractions. With higher starting fractions of vegetation, the spectral changes as a result of a burn can be larger. Secondly, the mean values of burned fraction (shown in boxplots) vary only very slightly, deviating up to 0.1 from the average of the satellites. This is in contrast with the extreme sample results (range shown in gray), which can deviate up to 0.5 from the average. We can conclude from this that the satellite sensor influence is relatively unimportant when looking at averaged values of sample combinations, but can vary wildly with individual sample results.

It is important to note that the satellites show a characteristic deviation from the mean. Typically, the Sentinel-2 satellites required higher values of burned fractions before detection, essentially being slightly less sensitive to spectral impacts of fire. The opposite is true for the MODIS instrument, which also had the highest deviations from the other instruments. Finally, Landsat 8 had the least skewed, as well as the lowest absolute, deviation from the mean. However, the sensitivity to spectral changes does not directly translate to burned area sensitivity. The large differences in pixel sizes between the instruments is, for example, another important factor.

For subsequent results, the differences between the satellite instruments are small when compared to the variation introduced by other parameters. Therefore, the data in this section are shown for the Landsat 8 satellite, as it has the most balanced detection characteristics. The results for Sentinel-2A, Sentinel-2B, and MODIS are shown in Appendix B.

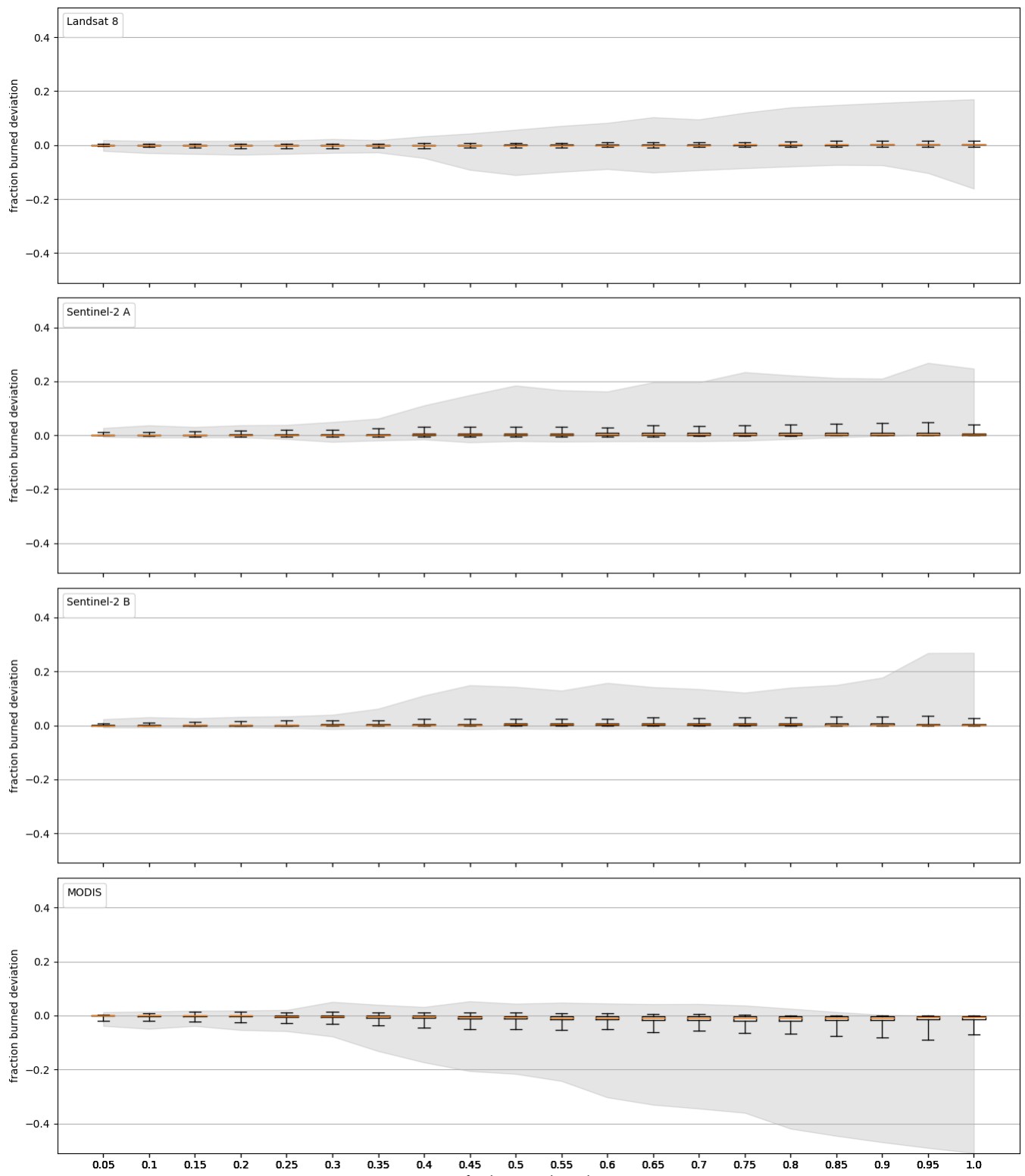

**Figure 3.** Deviation from mean of satellite instruments of burned fraction required for detection as a function of starting vegetation fraction. This compares model results with equal parameter settings and sample combinations when measured with the different satellite sensors. From top to bottom: Landsat 8, Sentinel-2 A, Sentinel-2 B, and the Moderate Resolution Imaging Spectroradiometer (MODIS). Total variation within the model results are shown in gray, while distribution of the expected (mean) values are shown in boxplots.

### 3.2. dNBR Threshold Influence

Detectability of a sample pixel was dependent on the detection threshold. When the dNBR threshold increases, a stronger burn spectral signal is required for detection to be reached. As a result, higher fractions of burned area were required, and the number of pixels that were undetectable increased (Figure 4).

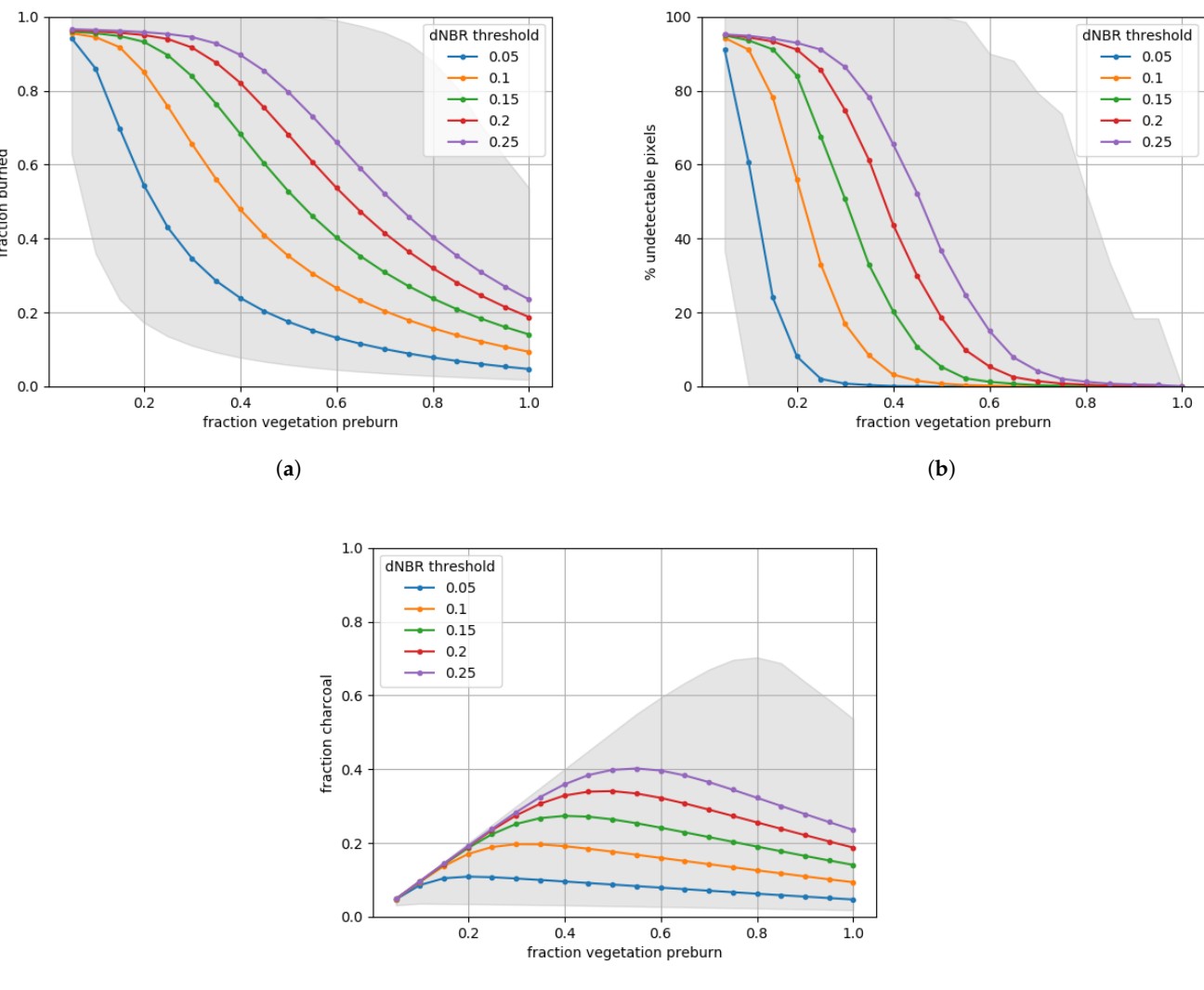

(**a**)

(**b**)

(**c**)

**Figure 4.** Park-wide detectability results for a pure burn scenario (where loss of vegetation fraction results in equal increase in charcoal fraction (Δchar = 1), Landsat 8. The mean values are shown in colored lines, with total model result variation shown in gray. (**a**) Fraction of vegetation burned at detection. (**b**) Percentage of undetectable pixels. (**c**) Charcoal contribution on detection.

In that sense, lower thresholds would be more desirable. However, this also increases the detection of non-burn-related changes in the spectral signal. For example, removal of vegetation changes the spectral signal in a similar way to a burn, as bare soil or rock typically has a lower NBR. We modeled this by setting Δchar to zero, where the removal of vegetation is compensated only by increases in substrate contribution ($f_g$). The resulting detection for this model run is shown in Figure 5.

Comparing the results of Figures 4 with 5, the conclusion can be taken that on average for the park, a vegetation removal scenario required higher levels of vegetation loss compared to a pure burn scenario. In addition, the percentage of undetectable pixels was higher. However, there seems to be no objectively optimal setting for the dNBR threshold.

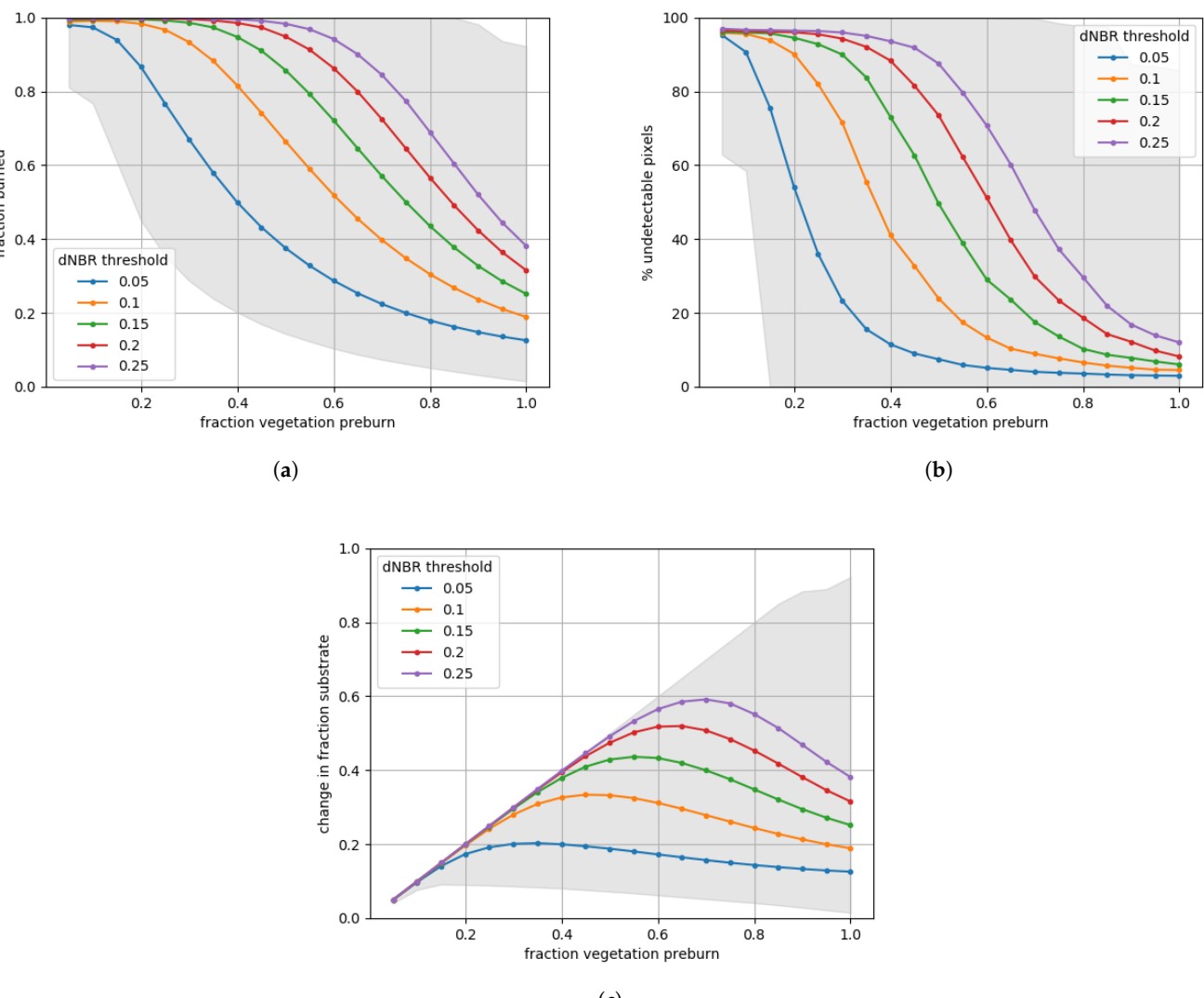

**Figure 5.** Park-wide detectability results for a vegetation removal scenario without fire (Δchar = 0), Landsat 8. The mean values are shown in colored lines, with total model result variation shown in gray. (**a**) Fraction of vegetation burned at detection. (**b**) Percentage of undetectable pixels. (**c**) Increase in substrate fraction on detection.

### 3.3. Park-Wide Detectability

Assessing the average number of undetectable pixels as a function of dNBR threshold and Δchar parameters (Figure 6), some important results become apparent. We can expect a burn's spectral impact to be somewhere between a pure burn scenario and a vegetation removal scenario; thus, the variability in detectability introduced by the Δchar parameter indicates a possible range of results. Here, it is notable that a significant fraction of the modeled pixels were undetectable. At the common dNBR threshold of 0.15, we find that even in the best case scenario (a pure burn scenario: Δchar = 1), more than a quarter of pixels were undetectable.

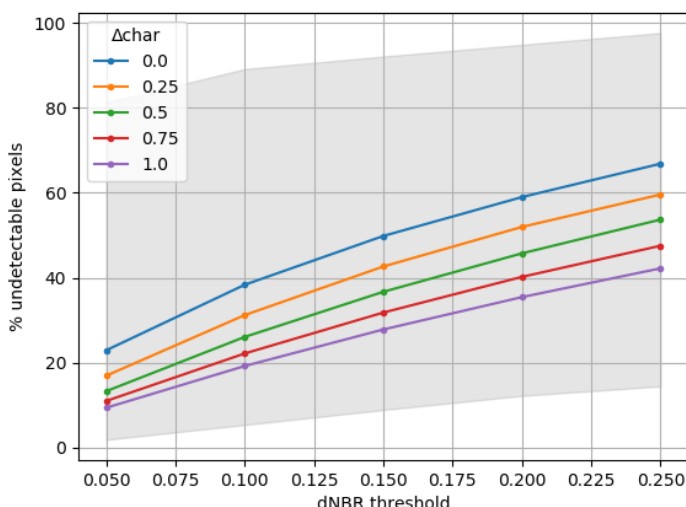

**Figure 6.** Percentage of undetectable burned pixels as a function of dNBR threshold and Δchar parameter settings, park-wide for Landsat 8. Mean values are shown in colored lines, with total model result variation shown in gray.

However, as shown previously, pixels with smaller burnable areas (low starting vegetation fractions) have lower detectability. The results for dNBR = 0.15 are shown in Figure 7. It shows that nearly all pixels with a starting vegetation fraction smaller than 0.2 were undetectable. For the best-case (pure burn) scenario, even at starting vegetation fractions of 0.4, 20% of burned pixels remained undetectable.

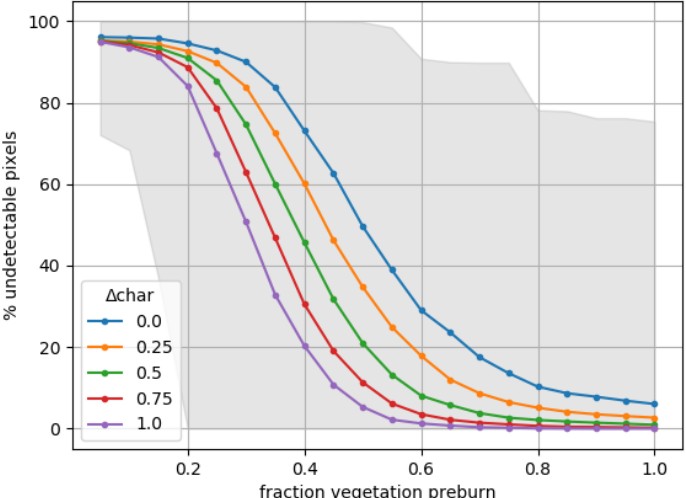

**Figure 7.** Percentage of undetectable pixels as a function of starting vegetation fraction and Δchar parameter setting, park-wide for Landsat 8, dNBR threshold = 0.15. Mean values are shown in colored lines, with total model result variation shown in gray.

Figure 8 shows the required vegetation cover area loss, before either detection or otherwise complete burn of the vegetation. These were calculated by multiplying fraction burned with the starting vegetation fractions, or alternatively by summing charcoal contribution and increases in substrate contribution. The mean values show that even pixels that started completely covered in vegetation ($f_{v,s} = 1$) required, on average, a vegetation loss at around 20% of the area of the pixel before detectability was reached. Pixels with a balanced starting contribution of vegetation ($f_{v,s} \approx 0.5$) resulted in the highest vegetation losses before detection, at 30 to 45% of the pixel's area.

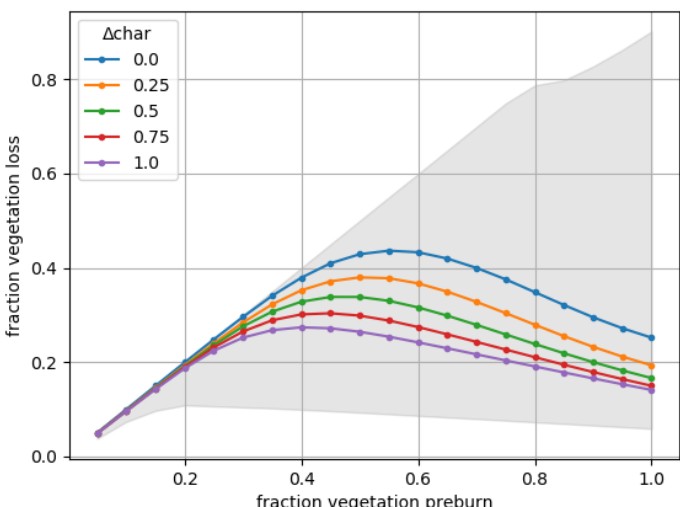

**Figure 8.** Vegetation loss results depending on starting vegetation fraction and Δchar parameter setting, park-wide for Landsat 8, dNBR threshold = 0.15. Mean values are shown in colored lines, with total model result variation shown in gray.

## 4. Discussion

There are notable uncertainties in our analysis, as denoted by the large range of possibilities in Figures 4–8. This range resulted from outliers within vegetation–geology groupings. These outliers represent burned pixels in which specific vegetation and geology spectral samples were combined. It is unlikely, yet not entirely impossible, that these combinations are fully representative for real-world combinations. The mean spectral responses of the vegetation–geology groupings showed much less variability, and those likely more realistically represent the real-world variability.

Our results indicate that more than 15% of an average pixel needs to burn before detection is reached, even in pixels with high vegetation covers. This is also illustrated by Randerson et al. [45], who estimated increases in burned area with 25 to 54% when including small fires that were not detected in the MODIS burned area product. The detection results are irrespective of pixel sizes; the estimated fraction of the pixel required to burn does not change with increasing resolution. However, its implication for burned area estimates is dependent on spatial resolution. With smaller pixel sizes, the burned area omission errors should decrease. Small burns cover larger fractions of smaller pixels, and, as such, have increased detectability. This means that burned area detection will be easier when using Landsat 8 and Sentinel-2 compared to using MODIS, which has much larger pixel sizes. For example, Ramo et al. [14] and Glushkov et al. [46] found significant increases in burned area estimates when reassessing MODIS burned area products with the higher-resolution Sentinel-2 imagery. However, the results shown in this work indicate that some omission errors are expected to persist, even for these smaller pixel sizes.

Our results shown in Figure 8 indicate that the highest omission errors are to be expected in sparsely vegetated terrain, as such pixels require nearly complete vegetation loss before detection. This effect may be further exacerbated, as sparsely vegetated areas may be more conducive to smaller fires because of fuel limitations [47]. Additionally, in regions that show high short-term spectral changes unrelated to fire, the small burned areas may be more difficult to discern from non-fire-related spectral shifts. This includes desert spring environments, where, as a result, small burned area detection from satellites is especially limited [48].

As our analysis uses linear spectral mixture analysis, it assumes that the light reflected by the surface interacts with only one component of the pixel and that there is no spectral influence of potential neighboring pixels. While this is a common assumption in spectral

mixture analysis, it is also a large simplification of reality, especially in vegetated areas where multiple and nonlinear scattering are common [49].

There are other factors in burned areas that may further influence detectability but were not considered in our analysis. The vegetation community spectral samples were collected from AVIRIS and therefore represent the vegetation at a specific time of the year. In this case, the measurements were taken on 6 August, near the peak of the growing season [31]. The assumption made in this work is that the pre-fire and post-fire satellite measurements occur on the anniversary date, and thus phenological shifts are minimal. However, any changes in spectral reflectance due to phenological shifts, moisture content, or shadows may alter the detectability of the burned areas [50].

We also did not explicitly account for different fire severity levels in our study, and this can be a major influence on the detectability of burned areas [51,52]. In our spectral modeling experiment, the fire effects were modeled by partly or fully replacing vegetation cover by soil and charcoal cover. In low to moderate fire severity plots, vegetation is often scorched. We have not accounted for the possibility of scorched vegetation; however, its inclusion in burned pixels instead of vegetation replacement by charcoal would likely even further decrease detectability.

One method for improving detectability of pixels with lower vegetation fractions is proposed by Miller and Thode [50], in which they scale dNBR by the pre-fire NBR. This results in the so-called Relative differenced Normalized Burn Ratio (RdNBR). This method increases sensitivity of the dNBR when the pre-fire NBR is close to 0. This improves detectability of burned areas in sparsely vegetated terrain. As our results indicate a high dependence on the pre-fire NBR of the pixels on their detectability, the RdNBR metric may yield significant detection improvements and should be further evaluated for burned area mapping in heterogeneous and sparsely vegetated areas.

Burned areas that cover small fractions of a pixel are more likely to be found either as a result of small, isolated fires or near the edges of larger fires. One way of limiting commission errors in the latter is by using region-growing algorithms [53]. In this method, pixels at the core of a burned region are first classified using a strict classification threshold. Next, a more relaxed threshold is used in the vicinity of these core burned pixels. This assumes that changes in NBR in vicinity to core burned pixels are likely resulting from the same fire. It allows more leniency towards smaller burned areas near the fringe of a fire, while changes in NBR unrelated to fire are not included. This technique may be used to improve detectability of burn fringes, but is not suitable for detecting small isolated fires.

We used dNBR thresholds to delineate burned area. While the dNBR has often been used as a burned area discriminator (e.g., [15,54]), we acknowledge that spectral reflectance in individual bands and other spectral indices are also often used (e.g., [11,19]). While small differences may exist depending on which spectral discriminator is used for burned area, we think that there would be commonalities with some of the main findings of our work with regards to, for example, the effect of pre-fire vegetation cover, burn fraction, and threshold values on burned area detectability.

The results presented in this work are representative for Yellowstone National Park, the region where the spectral endmembers were obtained from or selected for. While the park contains varied types of vegetation and substrate, Table 5 shows that the dominant lithologies in the park are various volcanic rocks, in total representing around 60% of the surface cover. In addition, more than 80% of the park's vegetation is represented by coniferous forests. In other regions, especially those with very different landscape characteristics, burned area detectability may yield different results.

## 5. Conclusions

We present a viable method for assessing satellite response to burned area. Using spectral data of vegetation, substrates, and charcoal and combining them using linear spectral mixture analysis, it was possible to model pre- and post-fire environments in Yellowstone National Park.

Differences between the spectral sensitivity of the Landsat 8, Sentinel-2, and MODIS to burned area were small. The MODIS instrument is slightly more sensitive towards burn spectral characteristics, while the Sentinel-2 satellites are relatively less sensitive. The Landsat 8 sensor showed a balanced response to burned area. These small influences may lead to slightly higher commission errors for MODIS and relatively higher omission errors for Sentinel-2.

Our results indicated that significant fractions of a pixel's vegetation need to burn before detection is reached. In addition, a significant percentage of the modeled pixels would remain undetectable. At the widely used dNBR threshold of 0.15, the average park-wide results showed that around a quarter of the pixel needs to be burned before detection becomes possible, and more than a quarter of the burned pixels remained undetectable. These results show that detection of burned area using dNBR may be accompanied by substantial omission errors.

Our spectral sensitivity analysis is independent of spatial scale, yet the implications become larger for coarser resolution sensors such as MODIS. Our results thus further support the continued investments in ongoing large-scale burned area mapping efforts at resolutions around 20 to 30 m from Sentinel-2 and Landsat 8. Using instruments with higher spatial resolutions directly lowers omission errors. Such efforts are especially important to quantify burned area from small isolated fires that often burn small amounts of fuels.

**Author Contributions:** Conceptualization, S.V. and M.R.; methodology, M.R. and S.V.; software, M.R.; formal analysis, M.R.; investigation, M.R.; resources, M.R. and S.V.; data curation, M.R.; writing—original draft preparation, M.R.; writing—review and editing, S.V.; visualization, M.R.; supervision, S.V.; project administration, S.V.; funding acquisition, S.V. All authors have read and agreed to the published version of the manuscript.

**Funding:** S.V. acknowledges support from the Dutch Research Council through Vidi grant 016.Vidi.189.070 and from the European Research Council under the European Union's Horizon 2020 research and innovation programme (grant agreement No. 101000987).

**Data Availability Statement:** The data used in this study are publicly available. The substrate endmember spectral samples were obtained from the ECOSTRESS spectral library [32]. The vegetation spectral samples were obtained from the USGS spectral library [31]. Landsat 8 response functions were obtained from the USGS website [39], Sentinel-2 response functions from the website of the European Space Agency [40], and MODIS response functions from the NASA website [41]. The surface geological and vegetation maps used to calculate sample abundances were obtained from the website of the USGS [43] and per email request from the Yellowstone National Park Service [27], respectively. The Python code written for this study, as well as map reclassification settings, can be found at https://github.com/matsriet/ynpburnedarea, accessed on 19 February 2022.

**Acknowledgments:** We thank three anonymous reviewers for their suggestions for improving the manuscript. Special thanks to Matt Jones from the University of East Anglia for checking the English language content of the manuscript.

**Conflicts of Interest:** The authors declare no conflict of interest.

## Abbreviations

The following abbreviations are used in this manuscript:

| | |
|---|---|
| USA or U.S. | United States of America |
| NBR | Normalized Burn Ratio |
| dNBR | differenced Normalized Burn Ratio |
| RdNBR | Relative differenced Normalized Burn Ratio |
| NIR | Near-infrared |
| SWIR | Shortwave infrared |
| SMA | Spectral mixture analysis |

| USGS | United States Geological Survey |
|---|---|
| AVIRIS | Airborne Visible/Infrared Imaging Spectrometer |
| NASA | National Aeronautics and Space Administration |
| ECOSTRESS | Ecosystem Spaceborne Thermal Radiometer Experiment on Space Station |
| MODIS | Moderate Resolution Imaging Spectroradiometer |

## Appendix A. Calculating Burned Fraction Directly

Given a target dNBR threshold, the endmember reflections, and cover contributions before the fire, it is possible to directly calculate the burned fraction needed for detection. This can be performed algebraically as the contribution fractions are all a linear function of $f_b$. The equations for $f_v$ (Equation (9)) and $f_c$ (Equation (10)) are already written as such, and the equation for $f_g$ can be rewritten using those:

$$f_g = 1 - (f_{v,s} \cdot (1 - f_b) + f_b \cdot f_{v,s} \cdot \Delta c) \tag{A1}$$

Leading to

$$f_g = f_b \cdot f_{v,s}(1 - \Delta c) - f_{v,s} + 1 \tag{A2}$$

The term $f_{v,s}(1 - \Delta c)$ is constant when calculating dNBR for a given pixel sample; thus, let us compact the equation by

$$k_g = f_{v,s}(1 - \Delta c) \tag{A3}$$

Yielding

$$f_g = f_b \cdot k_g - f_{v,s} + 1 \tag{A4}$$

Subsequently, Equation (6) can be rewritten to calculate NIR and SWIR reflectances as a function of $f_b$:

$$
\begin{aligned}
\rho_{p,NIR} &= \rho_{v,NIR} \cdot f_v + \rho_{g,NIR} \cdot f_g + \rho_{c,NIR} \cdot f_c \\
\rho_{p,NIR} &= \rho_{v,NIR} \cdot f_{v,s} \cdot (1 - f_b) \\
&\quad + \rho_{g,NIR} \cdot (f_b \cdot k_g - f_{v,s} + 1) \\
&\quad + \rho_{c,NIR} \cdot (f_b \cdot f_{v,s} \cdot \Delta c)
\end{aligned}
\tag{A5}
$$

$$
\begin{aligned}
\rho_{p,SWIR} &= \rho_{v,SWIR} \cdot f_v + \rho_{g,SWIR} \cdot f_g + \rho_{c,SWIR} \cdot f_c \\
\rho_{p,SWIR} &= \rho_{v,SWIR} \cdot f_{v,s} \cdot (1 - f_b) \\
&\quad + \rho_{g,SWIR} \cdot (f_b \cdot k_g - f_{v,s} + 1) \\
&\quad + \rho_{c,SWIR} \cdot (f_b \cdot f_{v,s} \cdot \Delta c)
\end{aligned}
\tag{A6}
$$

The postfire, or target, NBR given a certain dNBR threshold and pre-fire NBR can be calculated according to Equation (13), which gives the following inequality:

$$NBR_{p,s} - NBR_p \geq dNBR_{thresh} \tag{A7}$$

$$NBR_{p,s} - dNBR_{thresh} \geq NBR_p \tag{A8}$$

Inserting Equation (12) then yields

$$NBR_{p,s} - dNBR_{thresh} \geq \frac{\rho_{p,NIR} - \rho_{p,SWIR}}{\rho_{p,NIR} + \rho_{p,SWIR}} \tag{A9}$$

which is the inequality to solve to obtain the range of burned fraction for which it is detectable. $NBR_{p,s}$ can be calculated beforehand using Equation (8), and remains constant during the burn. The dNBR threshold ($dNBR_{thresh}$) is also constant. $\rho_{NIR,p}$ and $\rho_{p,SWIR}$ can be written as a function of $f_b$ and some parameters, as shown in Equations (A5) and (A6).

To solve the inequality, we first find the intercept between $NBR_{p,s} - dNBR_{thresh}$ and $\frac{\rho_{p,NIR} - \rho_{p,SWIR}}{\rho_{p,NIR} + \rho_{p,SWIR}}$. As we want to isolate $f_b$, it is useful to first multiply by the denominator:

$$(NBR_{p,s} - dNBR_{thresh}) \cdot (\rho_{p,NIR} + \rho_{p,SWIR}) = \rho_{p,NIR} - \rho_{p,SWIR} \tag{A10}$$

Inputting Equations (A5) and (A6):

$$\begin{aligned} \rho_{p,NIR} + \rho_{p,SWIR} &= (\rho_{v,NIR} + \rho_{v,SWIR}) \cdot f_{v,s} \cdot (1 - f_b) \\ &+ (\rho_{g,NIR} + \rho_{g,SWIR}) \cdot (f_b \cdot k_g - f_{v,s} + 1) \\ &+ (\rho_{c,NIR} + \rho_{c,SWIR}) \cdot f_b \cdot f_{v,s} \cdot \Delta c \end{aligned} \tag{A11}$$

$$\begin{aligned} \rho_{p,NIR} - \rho_{p,SWIR} &= (\rho_{v,NIR} - \rho_{v,SWIR}) \cdot f_{v,s} \cdot (1 - f_b) \\ &+ (\rho_{g,NIR} - \rho_{g,SWIR}) \cdot (f_b \cdot k_g - f_{v,s} + 1) \\ &+ (\rho_{c,NIR} - \rho_{c,SWIR}) \cdot f_b \cdot f_{v,s} \cdot \Delta c \end{aligned} \tag{A12}$$

For compactness sake, let us set the following shorthands for constants:

$$N = NBR_{p,s} - dNBR_{thresh} \tag{A13}$$

$$R_{x+} = \rho_{x,NIR} + \rho_{x,SWIR} \tag{A14}$$

$$R_{x-} = \rho_{x,NIR} - \rho_{x,SWIR} \tag{A15}$$

where $x$ can be substituted to indicate the pixel ($p$) or any of the endmembers ($v, g, c$). Yielding for Equations (A10)–(A12):

$$N \cdot R_{p+} = R_{p-} \tag{A16}$$

$$R_{p+} = R_{v+} \cdot f_{v,s} \cdot (1 - f_b) + R_{g+} \cdot (f_b \cdot k_g - f_{v,s} + 1) + R_{c+} \cdot f_b \cdot f_{v,s} \cdot \Delta c \tag{A17}$$

$$R_{p-} = R_{v-} \cdot f_{v,s} \cdot (1 - f_b) + R_{g-} \cdot (f_b \cdot k_g - f_{v,s} + 1) + R_{c-} \cdot f_b \cdot f_{v,s} \cdot \Delta c \tag{A18}$$

We can isolate $f_b$ in these equations:

$$R_{p+} = R_{v+} \cdot f_{v,s} - f_b \cdot R_{v+} \cdot f_{v,s} + f_b \cdot k_g \cdot R_{g+} + R_{g+} \cdot (1 - f_{v,s}) + f_b \cdot R_{c+} \cdot f_{v,s} \cdot \Delta c \tag{A19}$$

$$R_{p-} = R_{v-} \cdot f_{v,s} - f_b \cdot R_{v-} \cdot f_{v,s} + f_b \cdot k_g \cdot R_{g-} + R_{g-} \cdot (1 - f_{v,s}) + f_b \cdot R_{c-} \cdot f_{v,s} \cdot \Delta c \tag{A20}$$

and inserting them into Equation (A16) yields

$$\begin{aligned} &N \cdot R_{v+} \cdot f_{v,s} - f_b \cdot N \cdot R_{v+} \cdot f_{v,s} + f_b \cdot N \cdot k_g \cdot R_{g+} + N \cdot R_{g+} \cdot (1 - f_{v,s}) + f_b \cdot N \cdot R_{c+} \cdot f_{v,s} \cdot \Delta c \\ &= R_{v-} \cdot f_{v,s} - f_b \cdot R_{v-} \cdot f_{v,s} + f_b \cdot k_g \cdot R_{g-} + R_{g-} \cdot (1 - f_{v,s}) + f_b \cdot R_{c-} \cdot f_{v,s} \cdot \Delta c \end{aligned} \tag{A21}$$

Then, we can start solving for $f_b$:

$$\begin{aligned} &- f_b \cdot N \cdot R_{v+} \cdot f_{v,s} + f_b \cdot R_{v-} \cdot f_{v,s} + f_b \cdot N \cdot k_g \cdot R_{g+} - f_b \cdot k_g \cdot R_{g-} \\ &+ f_b \cdot N \cdot R_{c+} \cdot f_{v,s} \cdot \Delta c - f_b \cdot R_{c-} \cdot f_{v,s} \cdot \Delta c \\ &= R_{v-} \cdot f_{v,s} - N \cdot R_{v+} \cdot f_{v,s} + R_{g-} \cdot (1 - f_{v,s}) - N \cdot R_{g+} \cdot (1 - f_{v,s}) \end{aligned} \tag{A22}$$

$$\begin{aligned} &f_b \cdot (f_{v,s} \cdot (R_{v-} - N \cdot R_{v+}) + k_g \cdot (N \cdot R_{g+} - R_{g-}) + f_{v,s} \cdot \Delta c \cdot (N \cdot R_{c+} - R_{c-})) \\ &= f_{v,s} \cdot (R_{v-} - N \cdot R_{v+}) + (R_{g-} - N \cdot R_{g+}) \cdot (1 - f_{v,s}) \end{aligned} \tag{A23}$$

Yielding $f_{burned}$ at the intercept:

$$f_{b,intercept} = \frac{f_{v,s} \cdot (R_{v-} - N \cdot R_{v+}) + (R_{g-} - N \cdot R_{g+}) \cdot (1 - f_{v,s})}{f_{v,s} \cdot (R_{v-} - N \cdot R_{v+}) + k_g \cdot (N \cdot R_{g+} - R_{g-}) + f_{v,s} \cdot \Delta c \cdot (N \cdot R_{c+} - R_{c-})} \tag{A24}$$

The detectability status of the burn changes at $f_{b,intercept}$, with the burn being detectable at $f_b$ values higher or lower than this intercept. However, it turns out that $f_{b,intercept}$ always shows the lowest burned fraction at which the pixel becomes detectable. This is because $NBR_{p,s}$ is always higher than the intercept (as the dNBR threshold is always positive) and the asymptote cannot be between $0 \leq f_b \leq 1$. This is the case, because for the asymptote to be in this domain, the denominator in Equation (A10) would need to be 0. This will never happen for any combination of real endmember samples, as their reflectances would need to be equal to 0. Given that these reflectances are averages over a wavelength band, this does not occur. Thus, we can use Equation (A24) to directly calculate the burned fraction required for detection.

To check whether the outputs of our calculation are correct, every 1000 sample combination results were also calculated using a simple iteration method. The results for the geological unit Precambrian Gneiss and Schist is shown in Figure A1. The line has a stepped character due to the discretization used in the iterating method. This leads to deviations from calculation = iteration of, at maximum, the used step-size of $f_b$. None of the tested samples show a higher deviation than this step-size, indicating that the calculation is correct.

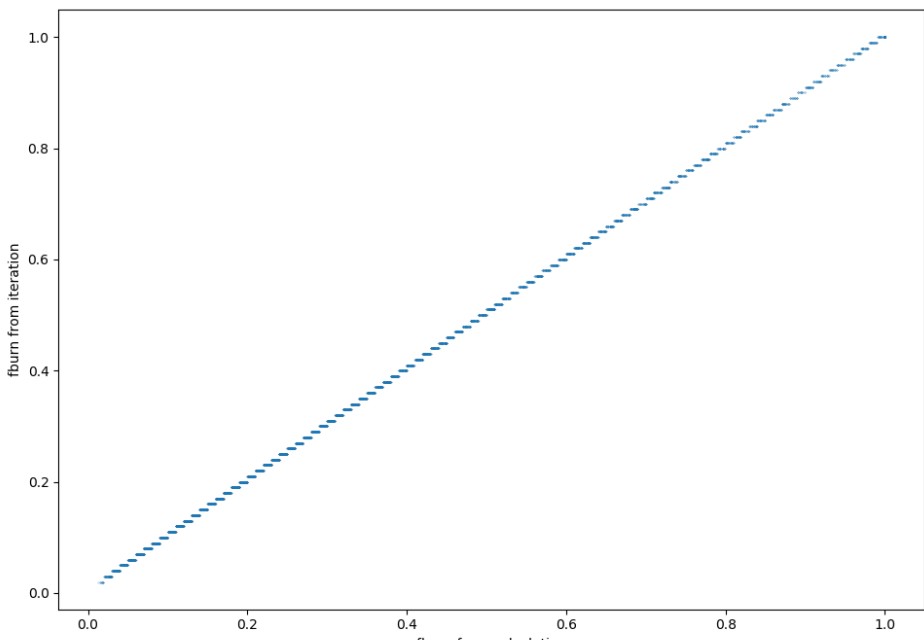

**Figure A1.** Model outputs tested with both iteration and direct calculation.

## Appendix B. Results for Other Satellite Sensors

Here, the results for the Sentinel-2A, Sentinel-2B, and the Moderate Resolution Imaging Spectroradiometer (MODIS) sensors are given.

*Appendix B.1. Results for Sentinel-2A*

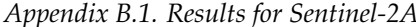

**Figure A2.** Park-wide detectability results for a pure burn scenario (where loss of vegetation fraction results in equal increase in charcoal fraction (Δchar = 1), Sentinel-2A. The mean values are shown in colored lines, with total model result variation shown in gray. To compare, the results for Landsat 8 are shown in black, with mean values shown by circles and total model result variation shown by dashed lines. (**a**) Fraction of vegetation burned at detection. (**b**) Percentage of undetectable pixels. (**c**) Charcoal contribution on detection.

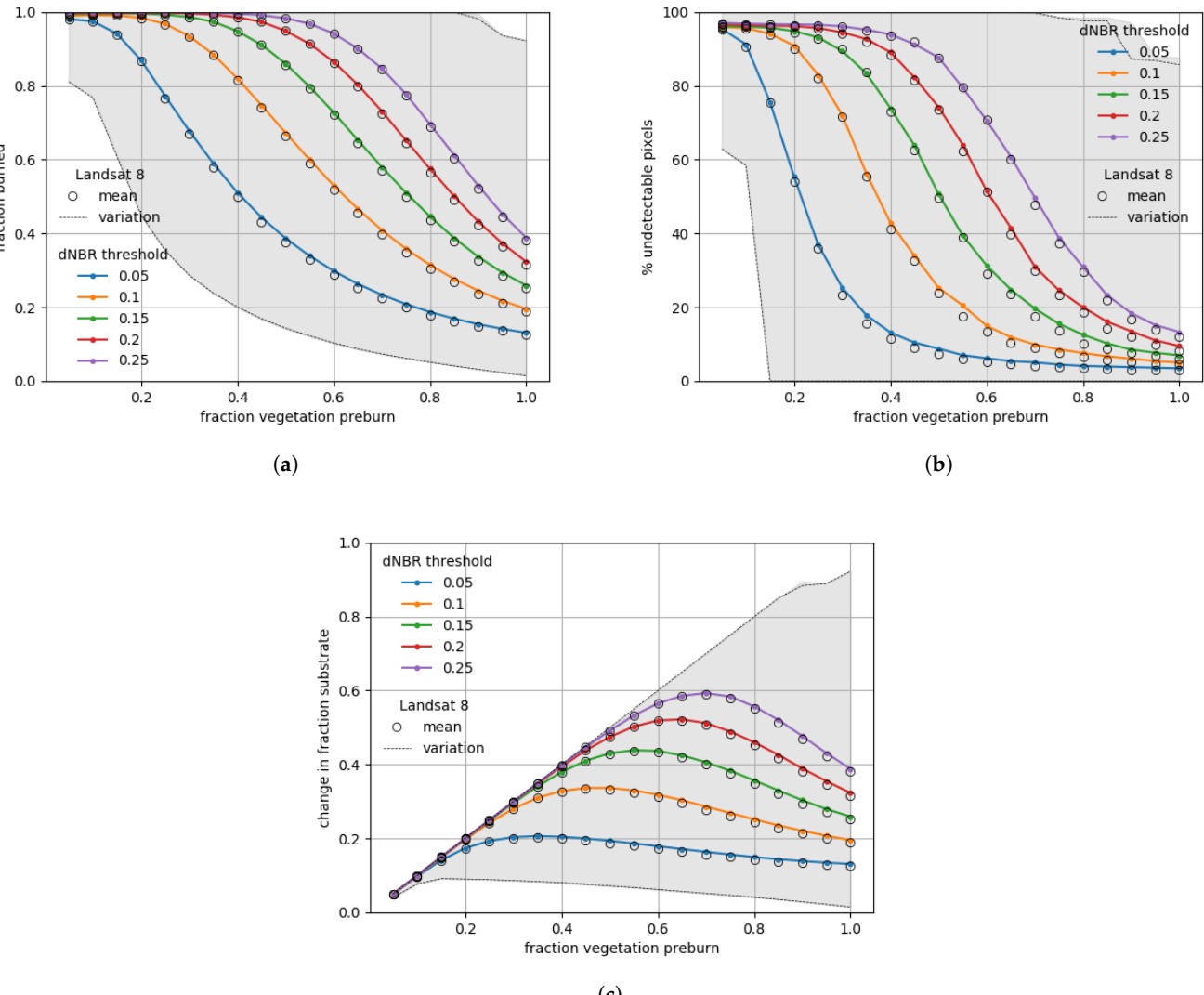

**Figure A3.** Park-wide detectability results for a vegetation removal scenario without fire (Δchar = 0), Sentinel-2A. The mean values are shown in colored lines, with total model result variation shown in gray. To compare, the results for Landsat 8 are shown in black, with mean values shown by circles and total model result variation shown by dashed lines. (**a**) Fraction of vegetation burned at detection. (**b**) Percentage of undetectable pixels. (**c**) Increase in substrate fraction on detection.

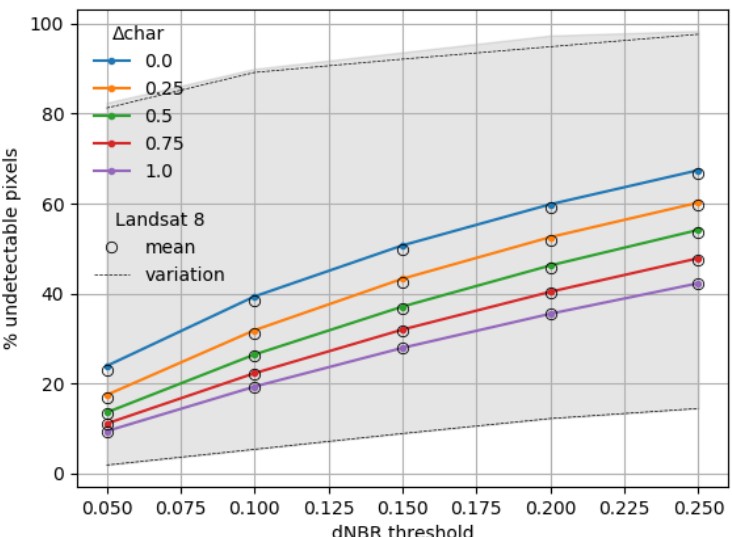

**Figure A4.** Percentage of undetectable burned pixels as a function of dNBR threshold and Δchar parameter settings, park-wide for Sentinel-2A. Mean values are shown in colored lines, with total model result variation shown in gray. To compare, the results for Landsat 8 are shown in black, with mean values shown by circles and total model result variation shown by dashed lines.

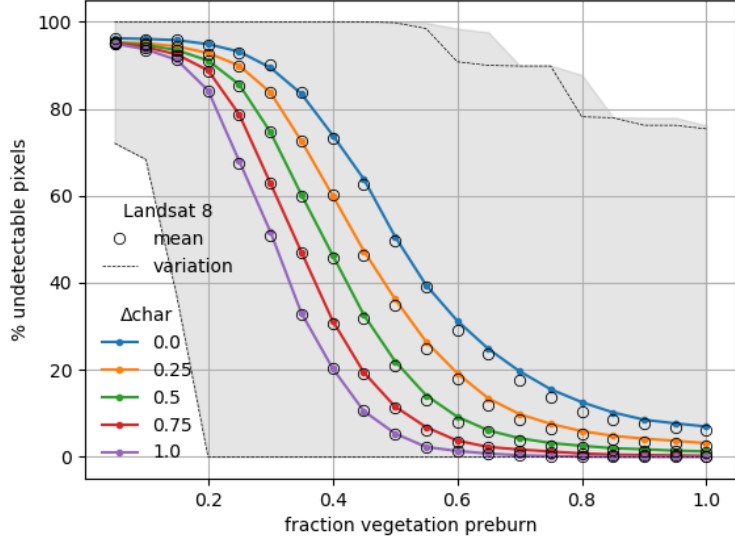

**Figure A5.** Percentage of undetectable pixels as a function of starting vegetation fraction and Δchar parameter setting, park-wide for Sentinel-2A, dNBR threshold = 0.15. Mean values are shown in colored lines, with total model result variation shown in gray. To compare, the results for Landsat 8 are shown in black, with mean values shown by circles and total model result variation shown by dashed lines.

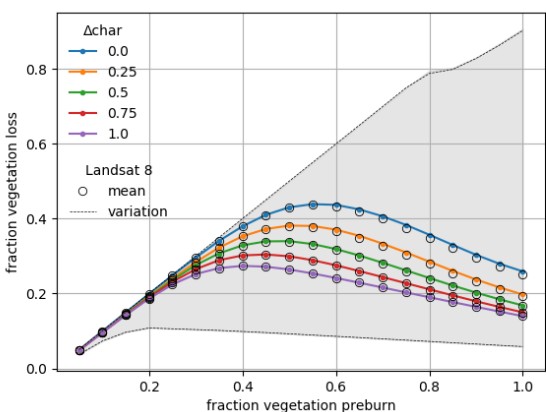

**Figure A6.** Vegetation loss results depending on starting vegetation fraction and Δchar parameter setting, park-wide for Sentinel-2A, dNBR threshold = 0.15. Mean values are shown in colored lines, with total model result variation shown in gray. To compare, the results for Landsat 8 are shown in black, with mean values shown by circles and total model result variation shown by dashed lines.

*Appendix B.2. Results for Sentinel-2B*

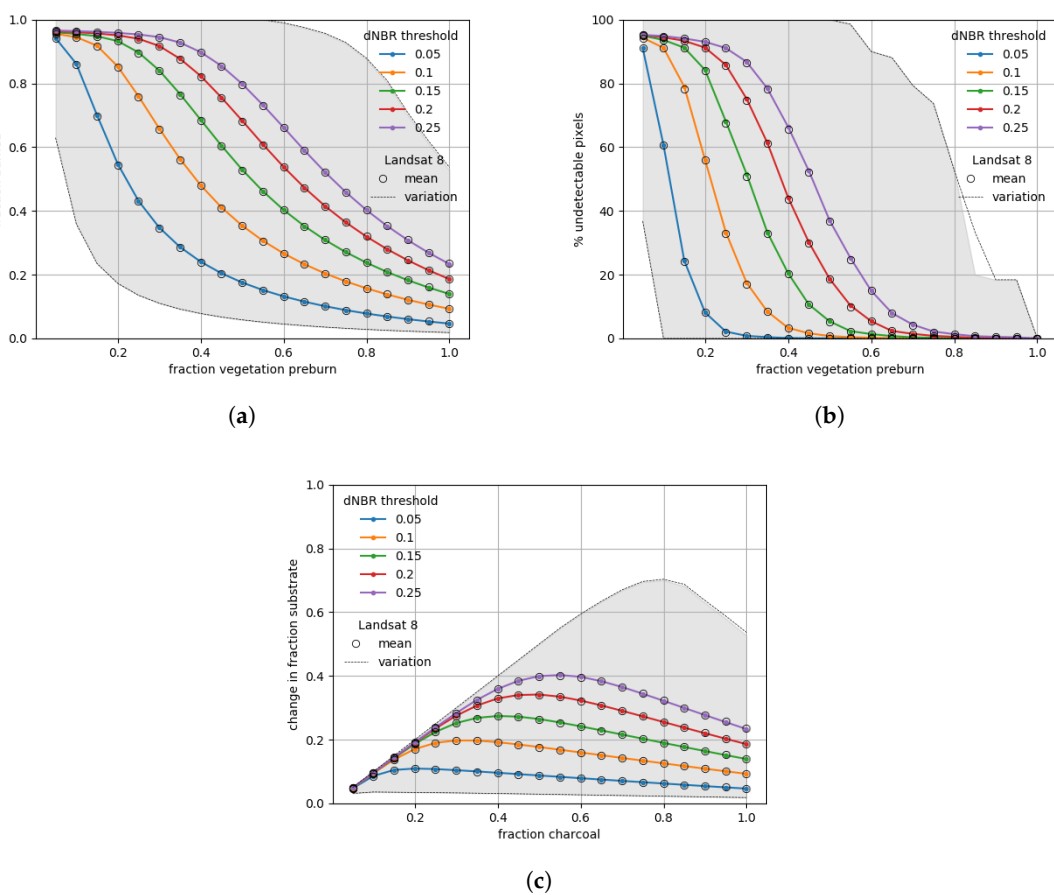

**Figure A7.** Park-wide detectability results for a pure burn scenario (where loss of vegetation fraction results in equal increase in charcoal fraction (Δchar = 1), Sentinel-2B. The mean values are shown in colored lines, with total model result variation shown in gray. To compare, the results for Landsat 8 are shown in black, with mean values shown by circles and total model result variation shown by dashed lines. (**a**) Fraction of vegetation burned at detection. (**b**) Percentage of undetectable pixels. (**c**) Charcoal contribution on detection.

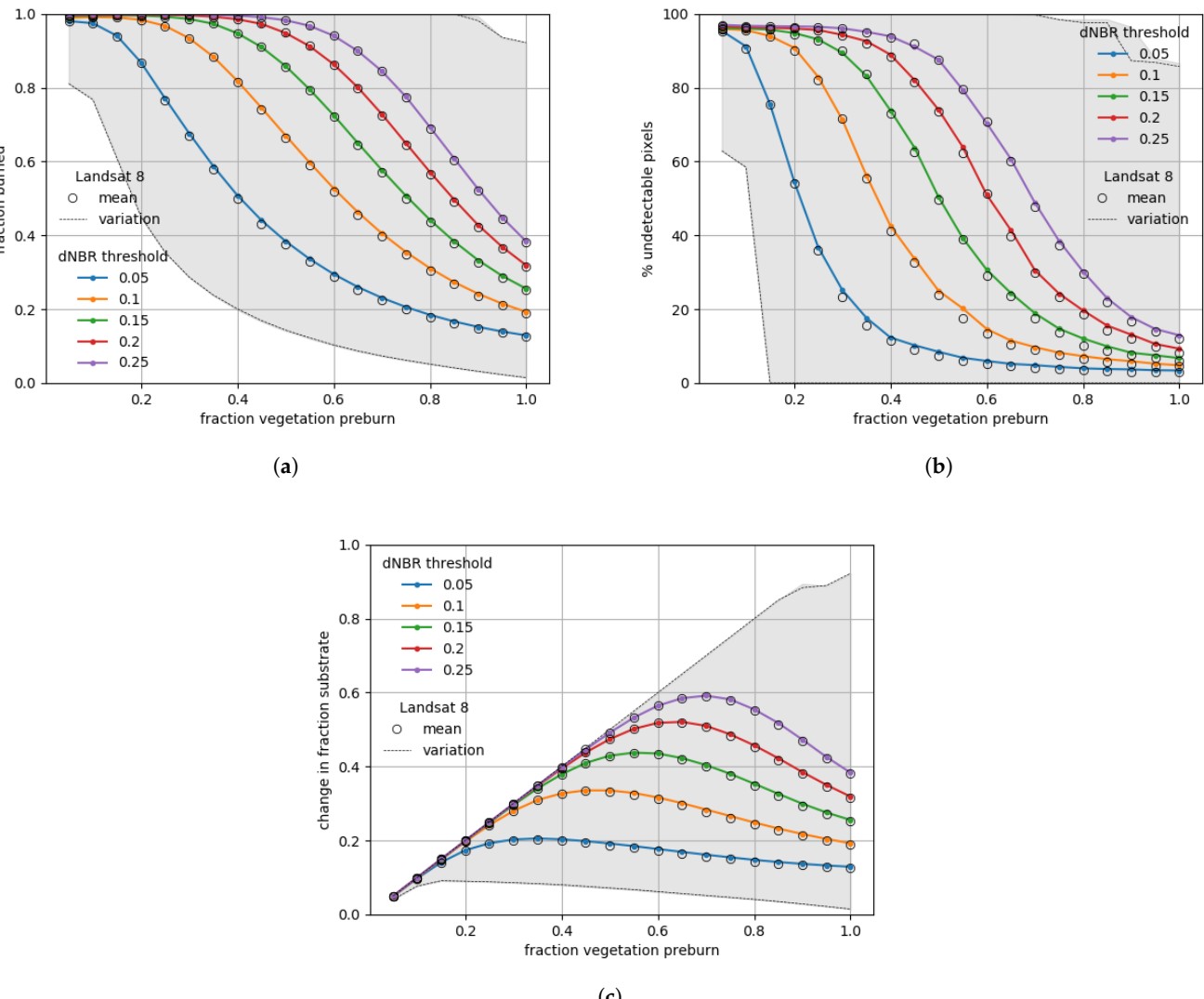

**Figure A8.** Park-wide detectability results for a vegetation removal scenario without fire (Δchar = 0), Sentinel-2B. The mean values are shown in colored lines, with total model result variation shown in gray. To compare, the results for Landsat 8 are shown in black, with mean values shown by circles and total model result variation shown by dashed lines. (**a**) Fraction of vegetation burned at detection. (**b**) Percentage of undetectable pixels. (**c**) Increase in substrate fraction on detection.

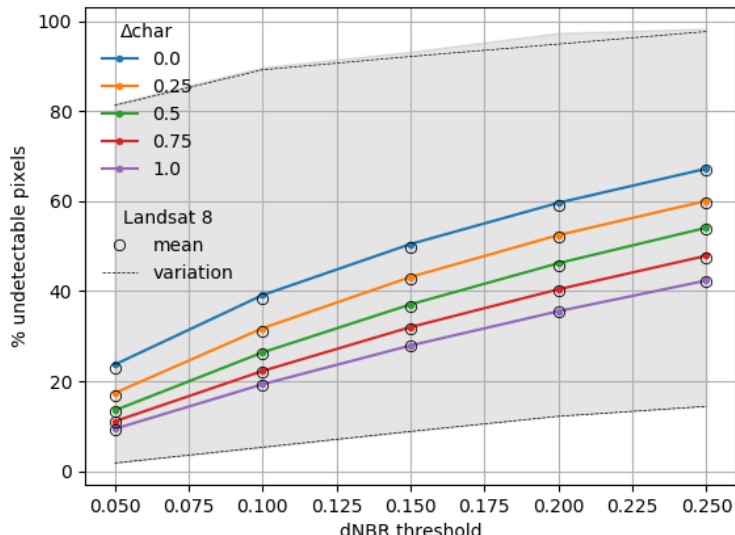

**Figure A9.** Percentage of undetectable burned pixels as a function of dNBR threshold and Δchar parameter settings, park-wide for Sentinel-2B. Mean values are shown in colored lines, with total model result variation shown in gray. To compare, the results for Landsat 8 are shown in black: with mean values shown by circles and total model result variation shown by dashed lines.

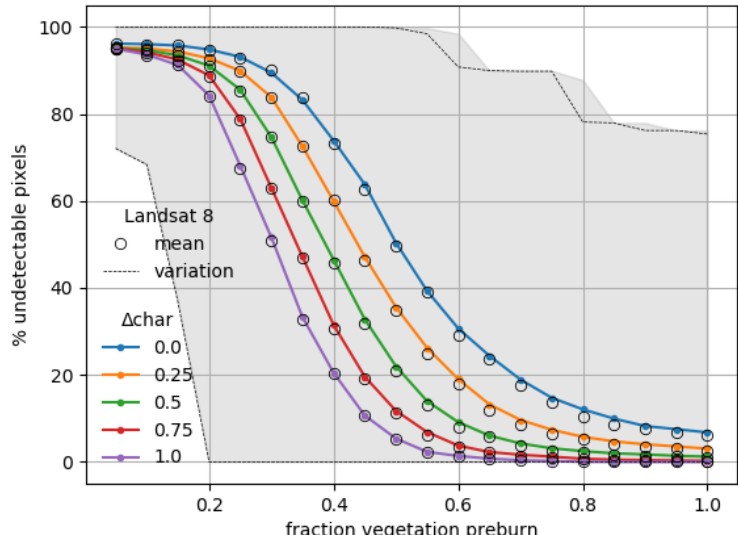

**Figure A10.** Percentage of undetectable pixels as a function of starting vegetation fraction and Δchar parameter setting, park-wide for Sentinel-2B, dNBR threshold = 0.15. Mean values are shown in colored lines, with total model result variation shown in gray. To compare, the results for Landsat 8 are shown in black, with mean values shown by circles and total model result variation shown by dashed lines.

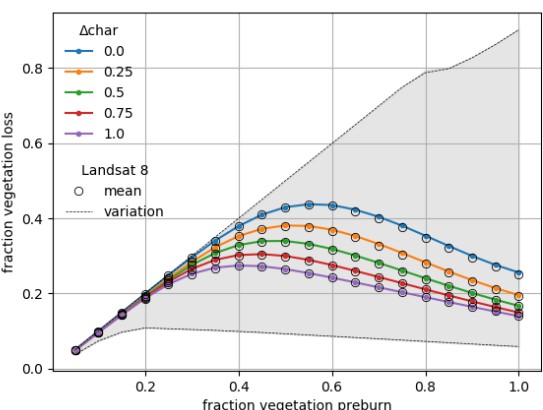

**Figure A11.** Vegetation loss results depending on starting vegetation fraction and Δchar parameter setting, park-wide for Sentinel-2B, dNBR threshold = 0.15. Mean values are shown in colored lines, with total model result variation shown in gray. To compare, the results for Landsat 8 are shown in black, with mean values shown by circles and total model result variation shown by dashed lines.

*Appendix B.3. Results for MODIS*

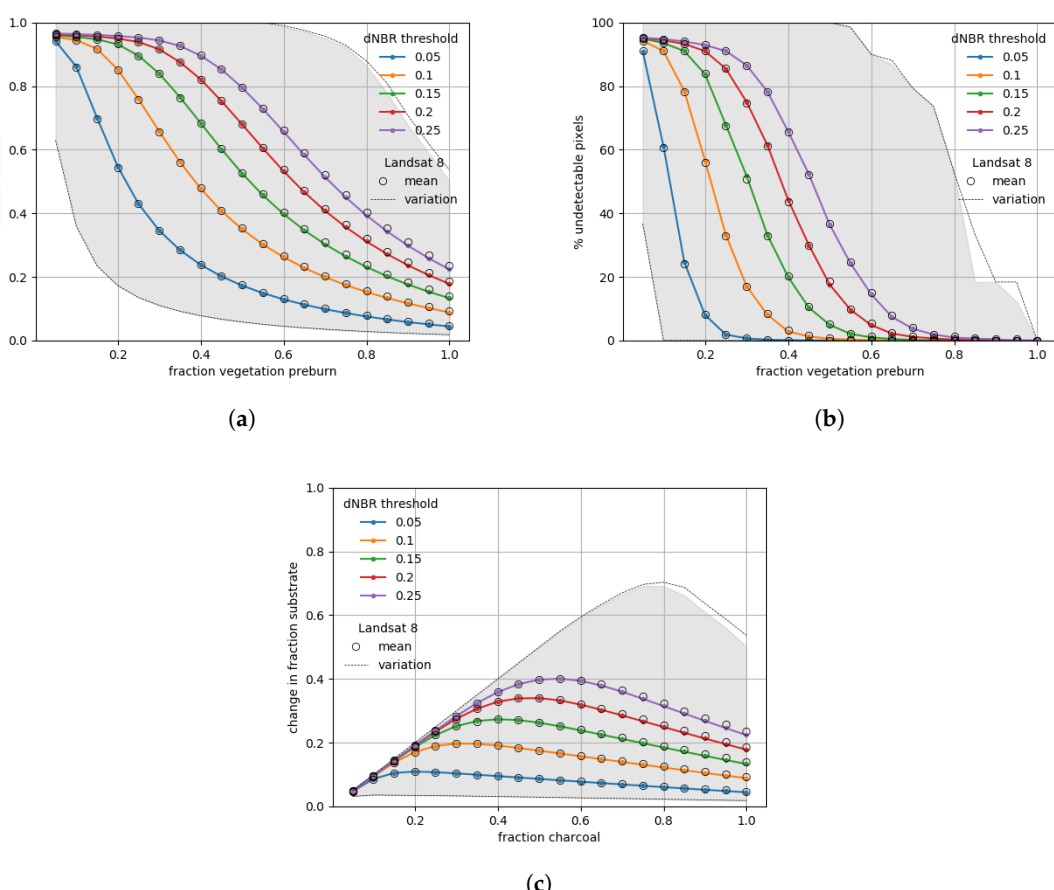

**Figure A12.** Park-wide detectability results for a pure burn scenario (where loss of vegetation fraction results in equal increase in charcoal fraction (Δchar = 1), for the Moderate Resolution Imaging Spectroradiometer (MODIS). The mean values are shown in colored lines, with total model result variation shown in gray. To compare, the results for Landsat 8 are shown in black, with mean values shown by circles and total model result variation shown by dashed lines. (**a**) Fraction of vegetation burned at detection. (**b**) Percentage of undetectable pixels. (**c**) Charcoal contribution on detection.

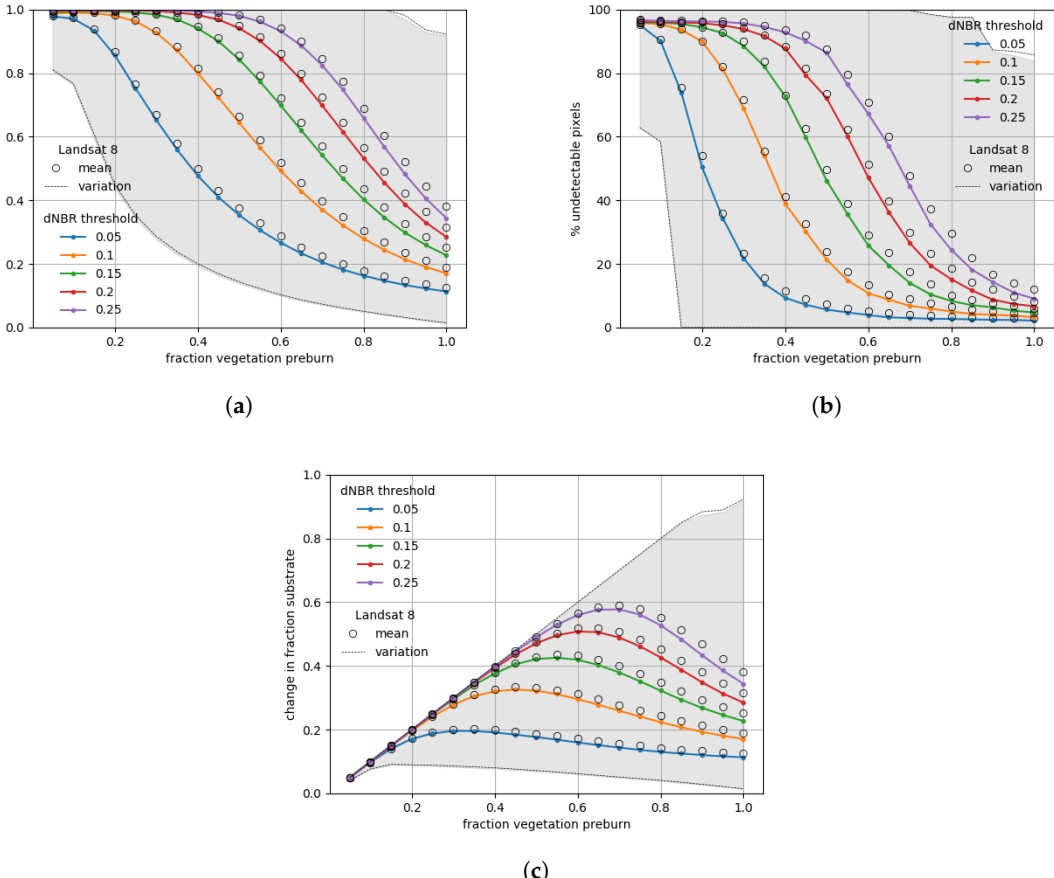

(**a**)  (**b**)

(**c**)

**Figure A13.** Park-wide detectability results for a vegetation removal scenario without fire (Δchar = 0), for the Moderate Resolution Imaging Spectroradiometer (MODIS). The mean values are shown in colored lines, with total model result variation shown in gray. To compare, the results for Landsat 8 are shown in black, with mean values shown by circles and total model result variation shown by dashed lines. (**a**) Fraction of vegetation burned at detection. (**b**) Percentage of undetectable pixels. (**c**) Increase in substrate fraction on detection.

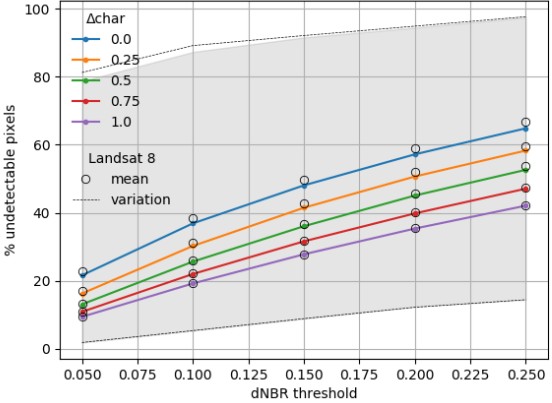

**Figure A14.** Percentage of undetectable burned pixels as a function of dNBR threshold and Δchar parameter settings, park-wide for the Moderate Resolution Imaging Spectroradiometer (MODIS). Mean values are shown in colored lines, with total model result variation shown in gray. To compare, the results for Landsat 8 are shown in black, with mean values shown by circles and total model result variation shown by dashed lines.

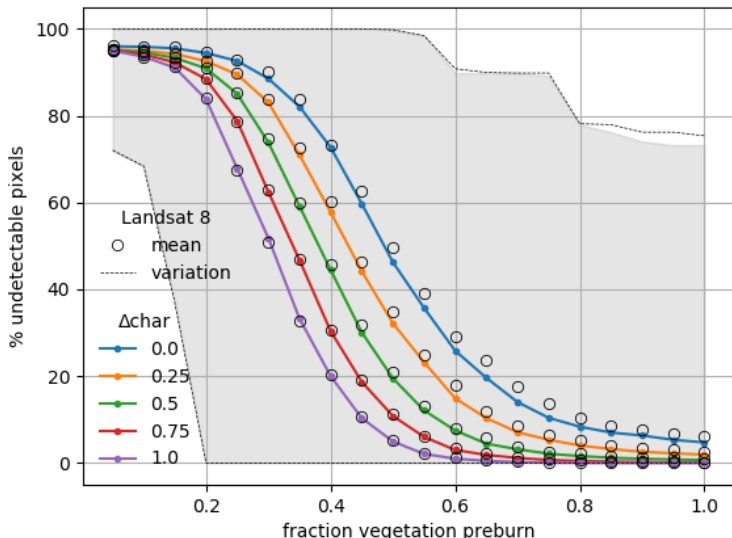

**Figure A15.** Percentage of undetectable pixels as a function of starting vegetation fraction and Δchar parameter setting, park-wide for the Moderate Resolution Imaging Spectroradiometer (MODIS), dNBR threshold = 0.15. Mean values are shown in colored lines, with total model result variation shown in gray. To compare, the results for Landsat 8 are shown in black, with mean values shown by circles and total model result variation shown by dashed lines.

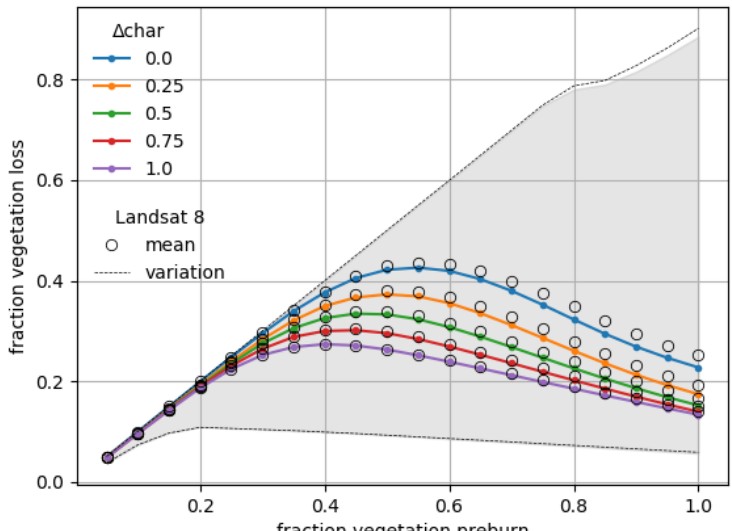

**Figure A16.** Vegetation loss results depending on starting vegetation fraction and Δchar parameter setting, park-wide for the Moderate Resolution Imaging Spectroradiometer (MODIS), dNBR threshold = 0.15. Mean values are shown in colored lines, with total model result variation shown in gray. To compare, the results for Landsat 8 are shown in black, with mean values shown by circles and total model result variation shown by dashed lines.

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
