# Peer review of "How Much of a Pixel Needs to Burn to Be Detected by Satellites? A Spectral Modeling Experiment Based on Ecosystem Data from Yellowstone National Park, USA"

_remotesensing, doi:10.3390/rs14092075_

Round 1

Reviewer 1 Report

Intercomparison of satellite sensors in burned area detectability is an important contribution to our understanding of remote sensing of fire. As a non-expert in sub pixel mixture analysis, I found some of the technical detail related to the SMA method a bit difficult to follow in parts. There are a couple of major concerns I've raised regarding the simple thresholding of dNBR, sometimes misleading comments of results relating to spatial resolution, and the real world application for fire mapping given the inherent fire behaviour characteristics not being equal across all pixels within a fire extent.  These concerns and other minor comments are outlined below. 

L60 – no satellite data used as input & L63 – endmember samples …obtained from spectral databases. & L75 ‘the available spectral data is complete and easily accessible’.

These references to the spectral database come before explaining about the USGS spectral library. Perhaps add (see section 2.2) at L63 to let the reader know there’s more information about that later.

L138-145: Does the sample preparation method use the spatial location of endmember points to correspond to the reflectance values in the imagery actually used? Or does the satellite instrument/timing of image capture vary between endmember analysis. Is there any standardisation in the satellite imagery used? Because there can be large differences in surface reflectance between seasons. So if your sample comes from different timing of imagery than what’s used pre and post-fire, that could introduce an extra source of variation.

Given SMA is widely used it surprised me that there are no citations in this section of the method. Has this approach been done before? If so it needs citations to help support the method used.

L192. Thresholding the dNBR might be ok for a localised study such as this, but it’s not a robust means of universally delineating burnt from unburnt. This should be acknowledged in the discussion.

Given you found a large effect of pre-fire vegetation cover, (which is related to why a simple threshold in dNBR is not adequate for delineating burnt from unburnt), you may get better results using the RdNBR, which has now been widely assessed as outperforming the simple dNBR. Thresholding the dNBR may not produce consistent results between fires or across the landscape due to the influence of pre-fire vegetation structure, soil type and vegetation moisture on the NBR See Miller and Thode, 2007 https://doi.org/10.1016/j.rse.2006.12.006  and  Kolden et al., 2015 https://doi.org/10.1071/WF15082 .  I’m not suggesting the method requires to be changed to RdNBR for this study, but at least these points could be acknowledged in the discussion.

Table 3. Add a column for spatial resolution of each satellite. For the wavelength ranges, add ‘NIR – range’ and ‘SWIR – range’, otherwise you could add a row at the top to identify band number and spectral range relating to NIR (merged columns in the top row) and SWIR, just to be explicit that the ranges columns relate to NIR vs SWIR in each case.

Table 4. ‘weighed’ should be ‘weighted’

L271. This result seems like it could be related to the different spatial resolutions of the different sensors? It may be misleading to say sentinel 2 is less sensitive to detecting fire, when the on-ground actual area that sentinel 2 can detect will be much smaller than Landsat and MODIS. I acknowledge you mention this in the discussion, saying smaller pixels inherently have increased detectability, but I think you need to refer to the pixel size differences in describing this result in the first place. Adding pixel size to Table 3 will help too.

L281. Given the variation in dNBR threshold is the cause of large variation in results for all endmembers, it may be worth the extra effort to refine this study by using the RdNBR. I predict a simple threshold in RdNBR would  be more robust to influence of pre-fire vegetation cover and would simplify your results.

L329:  The detection results are irrespective of pixel sizes; the estimated fraction of the pixel required to burn does not change with increasing resolution.

This statement seems to contrast with the result presented at L272-275 where you say Sentinel 2 required higher values of burnt fraction than MODIS.

L339 – again this result is likely related to the use of thresholding the dNBR and it’s limitations comparing across the landscape, (and may be overcome with use of the RdNBR) rather than an inherent limitation of satellite detectability.

L362. Good that you refer to Miller and Thode and the RdNBR. However, I don’t think it’s true that RdNBR increases commission error. Have you tested it? Where is the citation that gives the evidence of that statement. Many studies exist now that demonstrate RdNBR outperforms dNBR in accuracy statistics for fire extent and severity measures. The RdNBR deserves far greater consideration than to be dismissed with an unsupported statement such as this.

L382 Sentinel 2 can easily and practically be resampled to 10m resolution, as it’s only the SWIR bands that are 20m.

One important limitation of the method presented is not acknowledged is the common use in the literature of unburnt phenology offsets to adjust the dNBR (or RdNBR). This can greatly lower commission error, so it may further change your results and has nothing to do with inherent qualities of satellite sensor detectability.

This study considers the per pixel detectability, but does not consider the inherent spatial contagion of fire behaviour. Fires will often peter out to low severity at the edges of the fire boundary, and due to fire behaviour, pixels that are further inside the fire boundary are generally more likely to have complete coverage of the pixel area. Therefore, in practice, variation in subpixel detectability is more likely to affect the pixels at the fire boundary where low severity is more likely. The consequences of reduced satellite detectability of low severity at the fire boundary, in practice for operational fire mapping and to inform fire managers, is likely to be insignificant. This should be discussed and acknowledged.

Reviewer 2 Report

The authors present a straightforward spectral mixture analysis modelling technique to try and assess the satellite detectability of sub-pixel burned areas.  

I am broadly in favor of this student led paper, it could be an interesting and useful contribution.  However, there are  a number of major issues that would need to be addressed, followed by re-review. 

Main issues

1) Why is the delta NBR used to assess detectability ?   The delta NBR is not used to operationally map burned areas from MODIS, Landsat, or Sentinel-2.  

In addition, the NBR is highly sensitive to soil type in the presence of charcoal, for example, see: Smith, A. M., Eitel, J. U., & Hudak, A. T. (2010). Spectral analysis of charcoal on soils: Implicationsfor wildland fire severity mapping methods. International Journal of Wildland Fire19(7), 976-983. [ Cited > 50 times]

In addition, the NBR can sometimes be only poorly related to burn severity, for example see: French, N. H., Kasischke, E. S., Hall, R. J., Murphy, K. A., Verbyla, D. L., Hoy, E. E., and Allen, J. L., 2008, Using Landsat data to assess fire and burn severity in the North American boreal forest region: an overview and summary of results. International Journal of Wildland Fire, 17(4), 443-462) [Cited > 290 times]

Why not *also* show the detectability with respect to a single band such as the NIR that is used in most operational burned area mapping algorithms ?  

2) The combustion completeness is known to be influential in determining the ability to detect sub-pixel burns (not just the sub-pixel area burned).  This is not considered in the current version of the author’s paper but should be.

Please see the following 2 papers that discuss, and model,  combustion completeness and sub-pixel area burned effects:

Roy, D. P., Huang, H., Boschetti, L., Giglio, L., Yan, L., Zhang, H. H., & Li, Z. (2019). Landsat-8 and Sentinel-2 burned area mapping-A combined sensor multi-temporal change detection approach. Remote Sensing of Environment231, 111254. [ Cited > 95 times ]

Roy, D. P., & Landmann, T. (2005). Characterizing the surface heterogeneity of fire effects using multi‐temporal reflective wavelength data. International Journal of Remote Sensing26(19), 4197-4218. [Cited 110 times] 

3) Only three charcoal endmember samples were used – why is only 3 appropriate ?  At satellite resolution (even Sentinel-2 10 m resolution) burned areas are composed of black charcoal and white ash not just charcoal ?    How different are the 3 charcoal spectra ? Have other studies used more than 3 charcoal spectra ?  Are the results sensitive to different charcoal spectra ?

4) The three sensors (Landsat-8, Sentinel-2A/B, MODIS) do not provide noise-free reflectance - this is well established. Why is reflectance noise not modelled ?

5) The three sensors (Landsat-8, Sentinel-2A/B, MODIS) have different spatial resolutions – will the detectability vary with respect to sensor ?  Do the sensor spatial resolutions change across scan and track – will this influence the detectability ?  

6) Will shadows and soil moisture variations reduce the detectability ?

7) Is multiple scattering an issue for the mixture analysis modelling ?   

Line 36 – This is only part of the story;  see major comment 2.

Line 45 – Use a journal reference for the NBR.

Line 56 – Define “Fire Severity” and clarify if this is different to “Burn Severity”.

Line 71 – Unclear why stating that Yellowstone is the oldest national park in the world is relevant, here, and regardless Bogd Khan Uul is likely much older.

Line 92 – Introduce this better and clarify why so few vegetation spectra were used compared to the soil/rock spectra.

Line 103 – Please justify and clarify why green and non-photosynthetic vegetation can be grouped together meaningfully.  Currently this seems incorrect as they have very different spectra.  

Section 2.2 – Specify what the sample intervals (in nm) of the different spectral libraries are (they are I suspect different) and specify what the final spectral range used in the analysis was (refer to later table).  Clarify how this is handled in the later equations.

Before Line 140 - Add journal references for Equations 2, 3, 4 and 5.

Equations 5-10  – The equation terms are not clearly defined, too hard to follow, also please use the conventional Greek symbol for reflectance.  

Line 180 –“ represents forest loss without charcoal input (analogous to deforestation)” what about grass systems ?  This is confusing.

Line 202 – “Therefore, of these 627 samples, only the minimal, mean, and maximal values of the results were saved”  - This is not a clear justification, and seems nonsensical when on Line 212 it is stated that “The direct calculation is faster and more precise, allowing more samples to be processed within a reasonable amount of time.” and in the abstract that “50 million endmember combinations were assessed on burn detectability using the widely used”.

I stopped reviewing at this point given the major issues.  Hope that teh authors can address them.

Reviewer 3 Report

The topic is interesting, however, some important items are not clear and need being clarified.

1) The novelty should be explicitly clarified, especially with respect to the methodology.

2) Besides performances comparison in Fig.3, the detection results by Landsat 8, Sentinel-2 and MODIS should be added.

3)Can the conclusion drawn in this paper could be held in other areas? If possible, add another study area.

Round 2

Reviewer 3 Report

The paper could be accepted after English is improved.
